# Fast particulate nitrate formation via $N_2O_5$ uptake aloft in winter Beijing

*Haichao Wang[1], Keding Lu[1]\*, Xiaorui Chen[1], Qindan Zhu[1, #], Zhijun Wu[1], Yusheng Wu[1], Kang Sun[2]*

[1]State Key Joint Laboratory of Environmental Simulation and Pollution Control, College of Environmental Sciences and Engineering, Peking University, Beijing, China

[2]China National Environmental Monitoring Centre, Beijing, China

[#]Now at the Department of Chemistry, University of California, Berkeley, CA 94720, USA

*Correspondence to: Keding Lu (k.lu@pku.edu.cn)

**Abstract.**

Particulate nitrate ($pNO_3^-$) is an important component of secondary aerosols in urban areas. Therefore, it is critical to explore its formation mechanism to assist with the planning of haze abatement strategies. Here we report vertical measurement of $NO_x$ and $O_3$ by in-situ instruments on a movable carriage on a tower during a winter heavy-haze episode (December 18 to 20, 2016) in urban Beijing, China. Based on the box model simulation at different height, we found that $pNO_3^-$ formation via $N_2O_5$ heterogeneous uptake was negligible at ground level due to $N_2O_5$ concentration of near zero controlling by high NO emission and NO concentration. In contrast, the contribution from $N_2O_5$ uptake was large at high altitudes (e.g., > 150 m), which was supported by the low total oxidant ($NO_2 + O_3$) level at high altitudes than that at ground level. Modeling results show the specific case that the nighttime integrated production of $pNO_3^-$ for the high-altitude air mass above urban Beijing was estimated to be 50 μg m$^{-3}$ and enhanced the surface-layer $pNO_3^-$ the next morning by 28 μg m$^{-3}$ through vertical mixing. Sensitivity tests suggested that the nocturnal $NO_x$ loss by $NO_3$-$N_2O_5$ chemistry was maximized once the $N_2O_5$ uptake coefficient was over $2 \times 10^{-3}$ on polluted days with $S_a$ was 3000 μm$^2$ cm$^{-3}$ in wintertime. The case study

provided a chance to highlight that $pNO_3^-$ formation via $N_2O_5$ heterogeneous
hydrolysis may be an important source of the particulate nitrate in the urban airshed
during wintertime.

## 1. Introduction

Winter particulate matter (PM) pollution events occur frequently in China and have
drawn widespread and sustained attention in recent years (Guo et al., 2014; Zhang et
al., 2015; Huang et al., 2014; Wang G et al., 2016). PM pollution reduces visibility
(Lei and Wuebbles, 2013) and has harmful effects on public health (Cao et al., 2012).
Particulate nitrate ($pNO_3^-$) is an important component of secondary inorganic aerosols
and contributes 15% – 40% of the $PM_{2.5}$ mass concentration in China (Sun et al., 2013,
2015a, 2015b; Chen et al., 2015; Zheng et al., 2015; Wen et al., 2015). The main
atmospheric pathways of nitrate formation are (1) the reaction of OH with $NO_2$ and (2)
$N_2O_5$ heterogeneous hydrolysis (Seinfeld and Pandis, 2006). The reaction of OH with
$NO_2$ is a daytime pathway, as OH is very low in concentration at night, and $N_2O_5$
uptake is a nighttime pathway, as $NO_3$ and $N_2O_5$ are easily photo-labile.
Particulate nitrate formation via $N_2O_5$ heterogeneous hydrolysis in summer was
proved efficient by ground-based observation in North China (Wang H et al., 2017b;
Wang Z et al., 2017), and found comparable to or even higher than the daytime
formation. Several studies showed that $N_2O_5$ hydrolysis is responsible for nocturnal
$pNO_3^-$ enhancement in summer Beijing (Pathak et al., 2009, 2011; Wang H et al.,
2017a). Although $pNO_3^-$ formation via $N_2O_5$ uptake is significant in summertime, the
importance of this pathway in wintertime is not well characterized. Many differences
in $N_2O_5$ chemistry exist between winter and summer. First, as the key precursor of
$NO_3$ and $N_2O_5$, $O_3$ has a much lower concentration in winter than in summer, owing
to the short daytime length and weak solar radiation. Second, colder temperature and
high $NO_2$ level favor the partitioning towards $N_2O_5$. Third, longer nighttime length in
winter makes $N_2O_5$ heterogeneous hydrolysis potentially more important in $pNO_3^-$
formation. Finally, the $N_2O_5$ uptake coefficient, as an important parameter in $N_2O_5$
heterogeneous hydrolysis, is likely very different from that in summer. This is because
the properties of aerosol particles (e.g., organic compounds, particulate nitrate, liquid
water contents, solubility, and viscosity) and meteorological conditions (e.g.,
temperature and relative humidity) differ between summer and winter (Chen et al.,
2015; Zhang et al., 2007). These effects would result in large variations in the $N_2O_5$
uptake coefficient (Wahner et al., 1998; Mentel et al., 1999; Kane et al., 2001;
Hallquist et al., 2003; Thornton et al., 2003; Bertram and Thornton, 2009; Tang et al.,
2012; Wagner et al., 2013; Grzinic et al., 2015). Several parameterization methods did
not have good performance in predicting $N_2O_5$ uptake coefficient accurately (Chang et
al., 2011; Chang et al., 2016).
In addition to the seasonal differences in $pNO_3^-$ formation via $N_2O_5$ uptake,
modeling and field studies showed high levels of $NO_3$ and $N_2O_5$ at high altitudes
within the nocturnal boundary layer (NBL), owing to the stratification of surface NO
and volatile organic compounds (VOCs) emissions, which lead to gradients in the loss
rates for these compounds as a function of altitude (e.g., Brown et al., 2007; Geyer
and Stutz, 2004; Stutz et al., 2004). The $pNO_3^-$ formation via $N_2O_5$ uptake contributes
to the gradients in the compounds percentage and size distribution of the particle
(Ferrero et al., 2010; 2012). On nights when $NO_3$ can't accumulate in the surface
layer owing to high NO emissions, $N_2O_5$ uptake can still be active aloft without NO
titration. The $N_2O_5$ uptake aloft leads to elevated $pNO_3^-$ formation in the upper layer
as well as effective $NO_x$ removal (Watson et al., 2002; S. G. Brown et al., 2006;
Lurmann et al., 2006; Pusede et al., 2016; Baasandorj et al., 2017). Field observations
at high altitude sites of Kleiner Feldberg, Germany (Crowley et al., 2010a); the
London British Telecommunications tower, UK (Benton et al., 2010); and Boulder,
CO, USA (Wagner et al., 2013) showed the elevated $N_2O_5$ concentrations aloft. Model
studies showed that $pNO_3^-$ varied at different heights and stressed the importance of
the heterogeneous formation mechanism (Kim et al., 2014; Ying, 2011; Su et al.,
2017). The mass fraction and concentration of $pNO_3^-$ in Beijing was reported higher
aloft (260 m) than at the ground level in Beijing (Chan et al., 2005; Sun et al., 2015b),
which was explained by favorable gas–particle partitioning aloft under lower
temperature conditions. Overall, the active nighttime chemistry in the upper level
plays an important role in surface PM pollution through mixing and dispersing within
the planet boundary layer (PBL) (Prabhakar et al., 2017), the pollution was even
worse in valley terrain regions coupled with adverse meteorological processes
(Baasandorj et al., 2017; Green et al., 2015).
To explore the possible sources of $pNO_3^-$ and the dependence of its formation on
altitude in wintertime in Beijing, we conducted vertical profile measurements of NO,
$NO_2$, and $O_3$ with a moving cabin at a tower platform in combination with
simultaneous ground measurements of more comprehensive parameters in urban
Beijing. A box model was used to investigate the reaction rate of $N_2O_5$ heterogeneous
hydrolysis and its impact on $pNO_3^-$ formation at different altitudes during a heavy
haze episode over urban Beijing. Additionally, the dependence of $NO_x$ removal and
$pNO_3^-$ formation on the $N_2O_5$ uptake coefficient was probed.

**2. Methods**
**2.1 Field measurement**
Ground measurements (15 m above the ground) were carried out on the campus of
Peking University (PKU; 39°59'21"N, 116°18'25"E) in Beijing, China. The vertical
measurements were conducted at the Institute of Atmospheric Physics (IAP), Chinese
Academy of Sciences (39°58'28"N, 116°22'16"E). The IAP site is within 4 km of the
PKU site. The locations of the PKU and IAP sites are shown in Fig. 1. At the PKU
site, dry-state mass concentration of $PM_{2.5}$ was measured using a TEOM 1400A
analyzer. $NO_x$ was measured via a chemiluminescence analyzer (Thermo Scientific,
TE-42i-TR), and $O_3$ was measured with a UV photometric $O_3$ analyzer (Thermo
Scientific, TE-49i). Dry-state particle number and size distribution (PNSD) was
measured from 0.01 to 0.7 µm with a Scanning Mobility Particle Sizer (SMPS; TSI
Inc. 3010). The instrumental parameters are summarized in Table S1. The data were
collected from December 16 to 22, 2016. Additionally, relative humidity (RH),
temperature (T), and wind direction and speed data were available during the
measurement period.
Vertical profile measurements were conducted from December 18 to 20, 2016, from
the tower-based platform (maximum height: 325 m) on the IAP campus. The $NO_x$ and
$O_3$ instruments were installed aboard a movable cabin on the tower. $NO_x$ and $O_3$ were
measured with two low-power, lightweight instruments (Model 405 nm and Model
106-L, 2B Technologies, USA). The Model 405 nm instrument measures $NO_2$ directly
based on the absorbance at 405 nm, and NO is measured by adding excess $O_3$
(conversion efficiency ~100%). The limit of detection of both NO and $NO_2$ is 1 part
per billion volume (ppbv), with an accuracy of 2 ppbv or 2% of the reading, and the
time resolution is 10 s (Birks et al., 2018). The Model 106-L instrument measures $O_3$
based on the absorbance at 254 nm, with a precision of 1 ppbv, or 2% of the reading,
and a limit of detection of 3 ppbv. $NO_x$ calibration was performed in the lab using a
gas calibrator (TE-146i, Thermo Electron, USA) associated with a NO standard (9.8
ppmv). The $O_3$ calibration was done with an $O_3$ calibrator (TE 49i-PS), which was
traceable to NIST (National Institute of Standards and Technology) standards annually.
Before the campaign, the $NO_x$ monitor was compared with a Cavity Attenuated Phase
Shift (CAPs) Particle Light Extinction Monitor, and the $O_3$ monitor was compared to
a commercial $O_3$ analyzer (TE-49i, Thermo Electron, USA). Good agreement was
found between the portable instruments and the conventional monitors. Height
information was retrieved via the observed atmospheric pressure measured by the
Model 405 nm instrument. The cabin ascended and descended at a rate of 10 m min$^{-1}$,
with a height limit of 260 m during the daytime and 240 m at night. The cabin stopped
after reaching the peak, and parameters were measured continually during the last 10
min of each cycle. One vertical cycle lasted for approximately 1 h. We measured two
cycles per day, one in the morning and the other in the evening. Six cycles were
measured in total during the campaign.

**2.2 Box model simulation**
A simple chemical mechanism (see R1–R5) was used in a box model to simulate the
nighttime $NO_3$ and $N_2O_5$ chemistry under NO free-air-mass conditions. Physical
mixing, dilution, deposition, or interruption during the transport of the air mass was
not considered in the base case, the physical influence to the model result will be
discussed in Sect. 3.4. Here, $f$ represents the $ClNO_2$ yield from $N_2O_5$ uptake.
Homogeneous hydrolysis of $N_2O_5$ and $NO_3$ heterogeneous uptake reaction were
neglected in this analysis because of the low level of absolute humidity and the
extremely low $NO_3$ concentration during wintertime (Brown and Stutz, 2012). The
corresponding rate constants of R1–R3 are those reported by Sander et al. (2011).

153          $NO_2 + O_3 \rightarrow NO_3 + O_2$                       (R1)

154          $NO_2 + NO_3 + M \rightarrow N_2O_5 + M$             (R2)

155          $N_2O_5 + M \rightarrow NO_2 + NO_3 + M$             (R3)

156          $NO_3 + VOCs \rightarrow Products$                  (R4)

157          $N_2O_5 + (H_2O \text{ or } Cl^-) \rightarrow (2\text{-}f)\,NO_3^- + f\,ClNO_2$     (R5)

Following the work of Wagner et al. (2013), the box model can be solved using six
equations (Eqs. 1–6). In the framework, $O_3$ is only lost via the reaction of $NO_2 + O_3$
and the change in the $O_3$ concentration can be expressed as Eq. 1. Eq. 2 can express
the losses of $NO_2$. The s(t) is between 0 and 1 and expressed as Eq. 5, the physical
meaning of s(t) is the ratio of $NO_3$ production which goes through $N_2O_5$ (either as
$N_2O_5$ or lost through uptake) to the total $NO_3$ production (Wagner et al., 2013). The
s(t) favors 0 when direct loss of $NO_3$ dominates and favors 1 when $N_2O_5$ uptake
dominates $NO_3$ loss. The model calculation has two steps. The first step is calculate
the mixing ratio of $NO_2$ and $O_3$ at time zero (herein designated as sunset). According
to Eqs. 1 and 2, the initial $NO_2$ (t=0) and $O_3$ (t=0) concentrations can then be
integrated backward in time starting with the measured concentrations of $NO_2$ and $O_3$
at each height. During the pollution period in winter in Beijing ($NO_2$ = 45 ppbv,
Temperature = 273 K, $S_a$ = 3000 $\mu m^2\ cm^{-3}$), the ratio of $N_2O_5$ to $NO_3$ is large enough,
i.e., 450. The pseudo-first-order loss rate of $N_2O_5$ heterogeneous uptake will be $1 \times 10^{-3}$
$s^{-1}$, with a $N_2O_5$ uptake coefficient of $5 \times 10^{-3}$. $N_2O_5$ uptake would contribute to the
$NO_3$ loss rate of 0.4 $s^{-1}$, which is much higher than the direct $NO_3$ loss through the
reaction of $NO_3$ with VOCs. Therefore, $N_2O_5$ uptake was proposed to be dominantly
responsible for the $NO_3$ loss and the initial s(t) was set to 1. Eq. 3 can describe the
sum concentration of $NO_3$ and $N_2O_5$. Assuming the equilibrium between $NO_3$ and
$N_2O_5$ is maintained after a certain period, based on the temperature-dependent
equilibrium rate constant ($k_{eq}$) and the modeled $NO_2$ at a certain time, Eq. 4 can be
used to determine the ratio of $N_2O_5$ to $NO_3$. Combined, Eqs. 1–4 allow for the
calculation of $NO_3$ and $N_2O_5$ concentrations considering stable $NO_3$ and $N_2O_5$ loss
rate constants ($k_{NO3}$ and $k_{N2O5}$, respectively). In the second step, a new s(t) was
calculated using the data from the first step (Eq. 5), new initial $NO_2$ and $O_3$
concentrations were then approximated, and $NO_3$ and $N_2O_5$ values were derived using
the same method as used in the first step. This process was repeated until the
difference between the two s(t) values was less than 0.005. The number of
adjustments to a new s(t) could not be calculated more than 10 times. Otherwise, the
calculating process would become non-convergent.
The modeled $N_2O_5$ concentrations and given $k_{N2O5}$ were then used to estimate
$pNO_3^-$ formation. The $HNO_3$ produced in R4 was not considered because many of the
products are organic nitrates (Brown and Stutz, 2012). Here, $k_{NO3}$ and $k_{N2O5}$ denote
the pseudo-first-order reaction rate constants of the total $NO_3$ reactivity caused by
ambient VOCs and $N_2O_5$ heterogeneous uptake, respectively. $k_{N2O5}$ is given in Eq. 6.
$S_a$ is the aerosol surface area, $C$ is the mean molecular speed of $N_2O_5$, and $\gamma_{N2O5}$ is the
$N_2O_5$ uptake coefficient. Sunset and sunrise times during the measurements were
16:55 and 07:30 (Chinese National Standard Time, CNST), and the running time of
the model set to 14.5 h from sunset to sunrise.
$$\frac{d\,[O_3]}{dt} = -k_{NO2+O3}[O_3][NO_2] \tag{1}$$

$$\frac{d\,[NO_2]}{dt} = -(1 + s(t)) \times k_{NO2+O3}[O_3][NO_2] \tag{2}$$

$$\frac{d\,[NO_3 + N_2O_5]}{dt} = k_{NO2+O3}[O_3][NO_2] - k_{N2O5}[N_2O_5] - k_{NO3}[NO_3] \tag{3}$$

$$\frac{[N_2O_5]}{[NO_3]} = k_{eq}[NO_2] \tag{4}$$

$$s(t) = \frac{\int_0^t k_{N2O5} \cdot [N_2O_5]dt + [N_2O_5]_t}{[O_3](0) - [O_3](t)} \tag{5}$$

$$k_{N2O5} = \frac{C \times S_a \times \gamma_{N2O5}}{4} \tag{6}$$

Dry-state $S_a$ at the PKU site was calculated based on the PNSD measurement, which was corrected to ambient (wet) $S_a$ for particle hygroscopicity via a growth factor (Liu et al., 2013). The uncertainty of the wet $S_a$ was estimated to be ~30%, which was associated with the error from dry PNSD measurement (~20%) and the growth factor (~20%). Nighttime averaged $S_a$ on the night of December 19 was about 3000 $\mu m^2$ $cm^{-3}$. PM measurements at the National Monitoring Sites proved this heavy haze pollution episode was a typical regional event (Fig. S1). Furthermore, synchronous study on the night of December 19, 2016, showed small variation in the vertical particle number concentration, with a boundary layer height of 340 m (Zhong et al., 2017). Overall, the $S_a$ measured at the PKU site can represent the urban Beijing conditions in horizontal and vertical scale (< 340 m). Although the PNSD information for particles larger than 0.7 $\mu m$ was not valid during the study period, the particles smaller than 0.7 $\mu m$ dominated more than 95% of the aerosol surface area in a subsequent pollution episode (01/01/2017 to 01/07/2017), and similar results also were reported in other studies (e.g., Crowley et al., 2010a; Wang et al., 2018). The possible lower bias of $S_a$ (5%) only led to a small overestimation of $N_2O_5$, i.e., 3.6%–4.2%, and an underestimation of $pNO_3^-$ of 0.2%–2.5% when $\gamma_{N2O5}$ varied from $1 \times 10^{-3}$ to 0.05.

The $N_2O_5$ uptake coefficient and $ClNO_2$ yield are key parameters in the estimation of $pNO_3^-$ formation (Thornton et al., 2010; Riedel et al., 2013; Wagner et al., 2013; Phillips et al., 2016). Wagner et al. (2013) shows the significant $pNO_3^-$ suppression of $N_2O_5$ uptake aloft in the wintertime in Denver, CO, USA, the uptake coefficient is 0.005 when the percentage of $pNO_3^-$ in the $PM_{2.5}$ mass concentration is 40%. As the proportion of nitrate in the particle mass concentration is similarly high in North China during wintertime (Sun et al., 2013, 2015a; Chen et al., 2015; Zheng et al., 2015; Wen et al., 2015), herein we fixed the uptake coefficient to 0.005 for the base model initial input. Because the model input of $ClNO_2$ yield only affects the value of

produced $pNO_3^-$ concentration and would not change the modeled $N_2O_5$ concentration,
we set the initial $f_{ClNO2}$ to zero. Previous work showed the averaged $k_{NO3}$ was 0.01 -
0.02 $s^{-1}$ in summer Beijing, with BVOCs contributing significantly (Wang H et al.,
2017a; Wang et al., 2018). The intensity of BVOCs emissions decreased in wintertime,
owing to the lower temperature and weak solar radiation, thus the $k_{NO3}$ should be
smaller than it is in summer. In this work, the model input $k_{NO3}$ was set to an arbitrary
and relatively high value of 0.02 $s^{-1}$ (equivalent to 0.2 ppbv isoprene + 40 parts per
trillion volume (pptv) monoterpene + 1.0 ppbv cis-2-butene), to constrain the impact
of $N_2O_5$ uptake in the model. A series of sensitivity tests was conducted to study the
uncertainties of the model simulation, and the detailed test sets are listed in Table 1,
included the test of $N_2O_5$ uptake coefficient and $k_{NO3}$. The $\gamma_{N2O5}$ sensitivity tests were
set from 0.001 to 0.05, and the $k_{NO3}$ tests were set to 0.001 $s^{-1}$, 0.01 $s^{-1}$, and 0.1 $s^{-1}$.
In the calculation of the particulate nitration formation by $N_2O_5$ uptake, an
assumption is that all soluble nitrate formed from $N_2O_5$ uptake goes to the particle
phase rather than the gas phase. The assumption would lead to an upper bias as the
degassing of gas phase $HNO_3$ from particulate nitrate. While in winter Beijing, the
mixing ratio of $NH_3$ was rich to tens of ppbv and always much higher than the
nocturnal gas phase $HNO_3$ (e.g., Liu et al., 2017). The high $NH_3$ suppressed the
degassing of particulate nitrate effectively. The measurement of gas phase $HNO_3$ and
$pNO_3^-$ in the surface layer of Beijing showed the soluble nitrate favor to particle phase
in winter, especially in polluted days. For example, the nocturnal ratio of $pNO_3^-$ to
total soluble nitrate was larger than 0.95 on average (Liu et al., 2017). Due to the low
temperature and high RH at high altitude, the ratio would increase and the degassing
of particulate nitrate is negligible.

**3. Results and discussion**
**3.1 Ground-based observations.**
A severe winter PM pollution event lasted from December 16 to 22, 2016, in Beijing.
Figure 2a shows the time series of $PM_{2.5}$ and other relevant parameters based on
ground measurements at the PKU site. The mass concentration of $PM_{2.5}$ began to
increase from December 16, reaching 480 μg m$^{-3}$ on December 20. A fast PM growth
event was captured, with an overall increment of 100 μg m$^{-3}$ on the night of December
19 (Fig. 2a). Throughout the pollution episode, the meteorological conditions
included high RH (50% ± 16%) and low temperature (2 ± 3 °C). The slow surface
wind speed (< 3 m s$^{-1}$) implied the atmosphere was stable (Fig. 2c, d). The daytime $O_3$
concentration was low, owing to high NO emission and weak solar radiation. After
sunset, $O_3$ at surface layer was rapidly titrated to zero by the elevated NO. The
presence of high NO concentrations would have strongly suppressed the concentration
of $NO_3$, further suppressing $N_2O_5$ near the ground. Figure 2b depicts the high amounts
of NO and $NO_2$ that were observed at ground level during the PM pollution episode,
suggesting that $pNO_3^-$ production via $N_2O_5$ uptake was not important near the ground
during the winter haze episode.

**3.2 Tower observations.**

Six vertical measurements of the total oxidants ($O_x = O_3 + NO_2$) below 50 m were
consistent with those measured at ground level and are shown in Fig. S2, confirming
that the two sites are comparable. On the night of December 20 (Fig. 3a), the $NO_2$ and
NO from 0 – 240 m were abundant and conservative around 21:00, with
concentrations of 80 ppbv and 100 ppbv, respectively. The $O_3$ concentrations
remained zero during the nighttime (Fig. 3b). The vertical profile on December 20
suggests that at least below 240 m, the $N_2O_5$ chemistry was not important, which is
consistent with the results at ground level as mentioned above. The vertical profile on
December 19 was different with that on December 20. Figure 4a shows the vertical
profiles around 21:00 on December 19; NO was abundant from the ground to 100 m,
then gradually decreased to zero from 100 m to 150 m, and remained at zero above
150 m. The observed $NO_2$ concentration was 85 ± 2 ppbv below 100 m, which
gradually decreased from 100 m to 150 m, and was 50 ± 2 ppbv from 150 m to 240 m.
The observed $O_3$ concentrations below 150 m were below the instrumental limit of

detection (Fig. 4b). Above 150 m, the $O_3$ concentration was $20 \pm 2$ ppbv, corresponding to zero NO concentration. With respect to $O_x$, the mixing ratio of $O_x$ was $85 \pm 2$ ppbv at lower altitudes, whereas the $O_x$ concentration at higher altitudes was 15 ppbv lower than that at lower altitudes (Fig. 4b). The $O_x$ missing from the higher altitude air mass indicated an additional nocturnal removal of $O_x$ aloft.

Figure 5 depicts the vertical profiles of $NO_x$, $O_3$, and $O_x$ at 09:30 on the morning of December 20, which have similar features to those observed at 21:00 on December 19. The vertical profiles suggested stratification still existed at that time. The amount of $O_x$ missing aloft in the morning increased to 25 ppbv at $240 - 260$ m, demonstrating that an additional 25 ppbv of $O_x$ was removed or converted to other compounds at higher altitudes than at the surface layer during the night from December 19 to 20. Figure S3 shows the vertical profiles of NO, $NO_2$, $O_3$, and $O_x$ at ~12:00 on December 18, when solar radiation was strong enough to mix the trace gases well in the vertical direction. $NO_x$ and $O_3$ were found to be well mixed indeed, with small variation from the ground level to 260 m.

**3.3 Particulate nitrate formation aloft.**

$N_2O_5$ uptake is one of the two most important pathways of ambient $NO_x$ loss and $pNO_3^-$ formation (Wagner et al., 2013; Stutz et al., 2010; Tsai et al., 2014). At high altitudes (e.g., > 150 m), $NO_3$ and $N_2O_5$ chemistry can be initiated in the co-presence of high $NO_2$ and significant $O_3$ levels. Therefore, $N_2O_5$ uptake could represent a plausible explanation for the $O_x$ observed missing at high altitude on the night of December 19. To explore this phenomenon, a time-step box model was used to simulate the $NO_3$ and $N_2O_5$ chemistry based on the observed vertical profiles of $NO_2$ and $O_3$ on the night of December 19.

In the base case, the average initial $NO_2$ and $O_3$ levels above 150 m at sunset were $61 \pm 3$ ppbv and $27 \pm 6$ ppbv, respectively. The measured $NO_2$ concentration at the PKU site at sunset (local time, 16:55) was 61 ppbv and showed good consistency with the model result. The modeled $N_2O_5$ concentration was zero below 150 m, as the high

level of NO made for rapidly consumption of the formed $NO_3$. In contrast, the
modeled $N_2O_5$ concentrations at 21:00 above 150 m were in the range of 400–600
pptv (Fig. 6a). The $pNO_3^-$ formation by $N_2O_5$ heterogeneous uptake from sunset to the
measurement time can be calculated using Eq. 7, which was significant of 24 μg m$^{-3}$
after sunset above 150 m. The $pNO_3^-$ formed in 4.5 hours was equivalent to 13 ppbv
$O_x$ loss and consistent with the observation (15 ppbv) (Fig. 6b). Where the 1.5:1
relationship between $O_x$ and $pNO_3^-$ was used to calculate the $O_x$ equivalence (S. S.
Brown et al., 2006).
$$\sum pNO_3^- = \int_0^t (2-f) \cdot k_{N2O5} \cdot [N_2O_5] dt \qquad (7)$$
The box model enabled the analysis of the integrated $pNO_3^-$ and $ClNO_2$ via $N_2O_5$
uptake throughout the night. As shown in Fig. 6c, the modeled integrated $pNO_3^-$ went
as high as 50 μg m$^{-3}$. The integrated $pNO_3^-$ at sunrise was equal to the loss of 27 ppbv
$O_x$, showing a good agreement with the observed $O_x$ missing (25 ppbv) aloft in the
morning hours. During the nighttime, the $pNO_3^-$ formed aloft via $N_2O_5$ uptake led to
the much higher particle nitrate concentration than that in the surface layer, which has
been reported in many field observations (Watson et al., 2002; S. G. Brown et al.,
2006; Lurmann et al., 2006; Ferrero et al., 2012; Sun et al., 2015b). The elevated
$pNO_3^-$ aloft was well dispersed through vertical mixing and enhanced the
surface-layer PM concentration; this phenomenon was also observed in previous
studies (Watson et al., 2002; S. G. Brown et al., 2006; Lurmann et al., 2006;
Prabhakar et al., 2017). Zhong et al. (2017) showed that the NBL and PBL both were
at 340 m from December 19 to 20, 2016 in Beijing. Daytime vertical downward
transportation was helpful in mixing the air mass within the PBL. Assuming the newly
formed $pNO_3^-$ aloft from 150 m to 340 m is 50 μg m$^{-3}$ during the nighttime and well
mixed within the PBL in the next morning, the $pNO_3^-$ enhancement at the surface
layer ($\Delta pNO_3^-$) can be simplified to the calculation in Eq. 8 as following:
$$\Delta pNO_3 = \frac{\int_0^{150} P(pNO_3) dH + \int_{150}^{340} P(pNO_3) dH}{340} \qquad (8)$$
Here, $P(pNO_3^-)$ is the integral production of $pNO_3^-$ and $H$ represents height. Owing

to high NO below 150 m, the $pNO_3^-$ formation via $N_2O_5$ uptake was zero. The enhancement of $pNO_3^-$ from 150 m to 340 m was calculated as 28 μg m$^{-3}$, which is in good agreement with the observed PM peak in the morning on December 20, with PM enhancement of ~60 μg m$^{-3}$. The result demonstrated that the nocturnal $N_2O_5$ uptake aloft and downward transportation were critical for understanding the PM growth process.

**3.4 Sensitivity studies.**

Previous studies have emphasized that the $N_2O_5$ uptake coefficient varies greatly (0.001 – 0.1) in different ambient conditions (Chang et al., 2011; Brown and Stutz, 2012; Wang H et al., 2016), which is the main source of uncertainties in this model. In the present research, sensitivity studies showed the modeled $N_2O_5$ concentration dropping from 3 ppbv to 60 pptv when the $N_2O_5$ uptake coefficients increased from 0.001 to 0.05 (Fig. 6a), as the $N_2O_5$ concentration is very sensitive to the loss from heterogeneous reactions. Compared to the base case, the accumulated $pNO_3^-$ was evidently lower at γ = 0.001 (44 μg m$^{-3}$). Low $N_2O_5$ uptake coefficients correspond to several types of aerosols, such as secondary organic aerosols (Gross et al., 2009), humic acids (Badger et al., 2006), and certain solid aerosols (Gross et al., 2008). When the $N_2O_5$ uptake coefficient increased from 0.005 to 0.05 (Fig. 6b, c), the increase in integral $pNO_3^-$ was negligible. The conversion capacity of $N_2O_5$ uptake to $pNO_3^-$ is maximized for a given, fixed value of the $ClNO_2$ yield. The conversion of $NO_x$ to $pNO_3^-$ was not limited by the $N_2O_5$ heterogeneous reaction rate, but limited by the formation of $NO_3$ via the reaction of $NO_2$ with $O_3$ during the polluted night.

For describing the nocturnal $NO_x$ removal capacity and $pNO_3^-$ formation via $NO_3$ and $N_2O_5$ chemistry, the overnight $NO_x$ loss efficiency (ε) was calculated using Eq. 9.

$$\varepsilon = \frac{\int_0^t 2 \times k_{N2O5} \cdot [N_2O_5] dt + \int_0^t k_{NO3} \cdot [NO_3] dt}{[NO_2](0)} \tag{9}$$

The case modeled typical winter haze pollution conditions in Beijing from sunset to sunrise, with the initial model values of $NO_2$ and $O_3$ set to 60 ppbv and 30 ppbv,

respectively. $S_a$ was set to 3000 μm$^2$ cm$^{-3}$, the ClNO$_2$ yield was zero, and $k_{NO3}$ was
0.02 s$^{-1}$. The reaction time was set to 14.5 h to represent an overnight period in
wintertime. The consumed NO$_3$ by the reaction with VOCs and N$_2$O$_5$ by uptake
reaction were regarded as NO$_x$ removal. Figure 7 shows the dependence of the
overnight NO$_x$ loss efficiency on the N$_2$O$_5$ uptake coefficient, as it varied from $1\times10^{-5}$
to 0.1. This is an increase from 20% to 56% with increasing $\gamma_{N2O5}$, which is similar to
the result addressed by Chang et al. (2011). The ceiling of overnight NO$_x$ loss via
NO$_3$-N$_2$O$_5$ chemistry was fixed when all the NO$_x$ loss was through N$_2$O$_5$ uptake in
polluted days, which is limited by the reaction time and the formation rate of NO$_3$
(R1). In this case, the N$_2$O$_5$ uptake contributed about 90% of the overnight NO$_x$ loss
(50.4%) when $\gamma_{N2O5}$ was equal to $2\times10^{-3}$. When $\gamma_{N2O5}$ was less than $2\times10^{-3}$, NO$_x$
removal increased rapidly with the increasing of $\gamma_{N2O5}$, which was defined as the
$\gamma_{N2O5}$-sensitive region. When $\gamma_{N2O5} \geq 2\times10^{-3}$, the contribution of N$_2$O$_5$ uptake to NO$_x$
loss was over 90% and became insensitive. This region was defined as the
$\gamma_{N2O5}$-insensitive region. According to Eqs. 3 and 5, high $S_a$, high NO$_x$, low $k_{NO3}$ or
low temperature allow the N$_2$O$_5$ uptake more easily located in the $\gamma_{N2O5}$-insensitive
region. Here, the critical value of the N$_2$O$_5$ uptake coefficient ($2\times10^{-3}$) was relatively
low compared with that recommended for the surface of mineral dust (0.013,
290-300K) (Crowley et al., 2010b; Tang et al., 2017) or determined in many field
experiments (e.g., S. S. Brown et al., 2006; 2009; Wagner et al., 2013; Morgan et al.,
2015; Phillips et al., 2016; Wang Z et al., 2017; Brown et al., 2016; Wang H et al.,
2017b; Wang X et al., 2017). This suggests that the NO$_x$ loss and pNO$_3^-$ formation by
N$_2$O$_5$ uptake were easily maximized in the pollution episode, and further worsening
the PM pollution.
In the base case, the modeled pNO$_3^-$ formation via N$_2$O$_5$ uptake was an upper limit
result, as the ClNO$_2$ yield was set to zero. High coal combustion emitted chloride into
the atmosphere of Beijing during the heating period (Sun et al., 2013), like the
emissions from power plants in north China. This enhanced anthropogenic chloride
provides abundant chloride-containing aerosols to form ClNO$_2$ via N$_2$O$_5$ uptake aloft,
implying that significant ClNO$_2$ formed in the upper layer of the NBL (Tham et al.,

2016; Wang Z et al., 2017). Assuming the ClNO$_2$ yield is the average value of 0.28

determined at high altitude in north China (Wang Z et al., 2017), the pNO$_3^-$ produced

throughout the night decreased 7 μg m$^{-3}$. The modeled formation of ClNO$_2$ aloft

throughout the night was 2.5 ppbv, which is comparable with the observation in North

China (Tham et al., 2016; Wang Z et al., 2017; Wang X et al., 2017). Since the

modeled pNO$_3^-$ formation is sensitive to the ClNO$_2$ yield, a higher yield would

increase the model uncertainty directly, hence probing the ClNO$_2$ yield is warranted in

future studies. As for NO$_3$ reactivity, Fig. 8 shows the sensitivity tests of the integral

pNO$_3^-$ formation for the whole night at $k_{NO3}$ values = 0.001 s$^{-1}$, 0.01 s$^{-1}$, 0.02 s$^{-1}$, and

0.05 s$^{-1}$. The integral pNO$_3^-$ formation decreased when $k_{NO3}$ varied from 0.001 s$^{-1}$ to

0.1 s$^{-1}$, but the variation ratio to the base case was within 5%. The result shows the

NO$_3$-N$_2$O$_5$ loss via NO$_3$ reaction with VOCs during the polluted wintertime was not

important, which may only lead to relatively small uncertainties in the integral pNO$_3^-$

formation calculation. Nevertheless, if N$_2$O$_5$ uptake was extremely low (e.g., γ$_{N2O5}$ <

10$^{-4}$), the uncertainty of NO$_3$ oxidation would increase significantly.

The uncertainty caused by the physical changes of the air masses were analyzed

from two folds, one is the dilution and the other is the mixing and exchange of the air

mass. With respect to the impact of the dilution process, it would decrease the mixing

ratio of NO$_2$, O$_3$, NO$_3$ and N$_2$O$_5$, and leads to a lower contribution to the particulate

nitrate formation. An additional loss process for trace gases with a lifetime of 24 h

was assumed for calculated species in the sensitivity test (Lu et al., 2012). The result

shows that the integrated production of particulate nitrate decreased 28% compared

with base case. With respect to the exchange and mixing of the air mass at high

altitude during nighttime in polluted winter, the stable atmospheric stratification was

featured with strong inversion (Zhong et al., 2017). The nocturnal atmosphere is

stable and layered, the upward mixing from the surface is minimized, and air masses

above the surface are less affected by nocturnal emissions (Wagner et al., 2013).

Nevertheless, the injection by warm combustion sources or the clean air mass can

affect the air mass in fact. If the warm combustion source emitted NO$_x$ into the air

mass after sunset, which would increase the mixing ratio of O$_x$, and restart the zero

time of the model. Accounting for the uncertainties from the mixing, sensitivities tests
of the box model to shorting the duration of 25%, the bias of the integrated $pNO_3^-$
throughout the night was small within 12% relative to base case. If the air mass was
affected by the clean air mass from the north, it would be featured with very low $NO_x$
and about 40 ppbv $O_3$ (background condition), which was not consistent with our
observation.

## 4. Conclusion

During the wintertime, ambient $O_3$ is often fully titrated at the ground level in urban
Beijing owing to its fast reaction with NO emissions. Consequently, the near-surface
air masses are chemically inert. Nevertheless, the chemical information of the air
masses at higher altitudes was indicative of a reactive layer above urban Beijing,
which potentially drives fast $pNO_3^-$ production via $N_2O_5$ uptake and contributes to the
surface PM mass concentration. In this study, we found a case to show evidence for
additional $O_x$ missing (25 ppbv) aloft throughout the night. Based on model
simulation, we found that the particulate nitrate formed above 150 m reached 50 μg
$m^{-3}$ and enhanced the surface level PM concentration significantly by 28 μg $m^{-3}$ with
downward mixing after break-up of the NBL in the morning.
Our result emphasized the importance of the heterogeneous chemistry aloft the city
through a case study. The model simulation also demonstrated that during the heavy
PM pollution period, the particulate nitrate formation capacity via $N_2O_5$ uptake was
easily maximized in the high altitude above urban Beijing, even with low $N_2O_5$ uptake
coefficient. This indicates that the mixing ratio of $NO_2$ aloft was directly linked to
nitrate formation, and reduction of $NO_x$ is helpful in decreasing nocturnal nitrate
formation. Overall, this study highlights the importance of the interplay between
chemical formation aloft and dynamic processes for probing the ground-level PM
pollution problem. In the future, direct observations of $N_2O_5$ and associated
parameters should be performed to explore the physical and chemical properties of
this overhead nighttime reaction layer and to reach a better understanding of the

winter haze formation.

*Acknowledgements.*

This work was supported by the National Natural Science Foundation of China (Grant No. 91544225, 41375124, 21522701, 41571130021), the National Key Technology Research and Development Program of the Ministry of Science and Technology of China (Grant No. 2014BAC21B01). The authors gratefully acknowledge the science team of Peking University for their general support, as well as the team running the tower platform, which enabled the vertical profile observations.

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

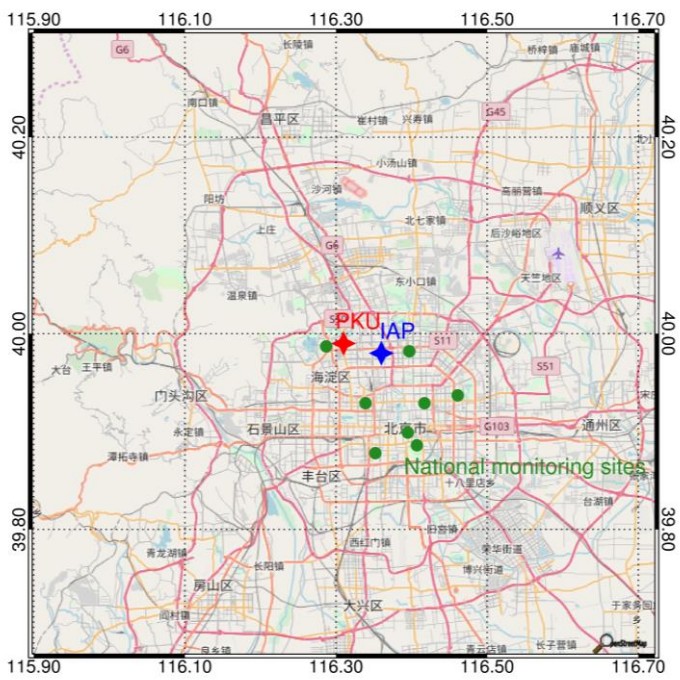


**Figure 1**. Location of the monitoring sites used in this study, including PKU (red diamond), IAP (blue diamond), and other National Monitoring Sites (green circles). Vertical profiles of $NO_x$ and $O_3$ were collected at a tower at the IAP. Measurements of particle number and size distribution (used to calculate $N_2O_5$ and particle nitrate formation) were collected from a ground site at PKU. Additional measurements on $PM_{2.5}$ concentrations were continuously measured at national monitoring sites throughout Beijing.

736

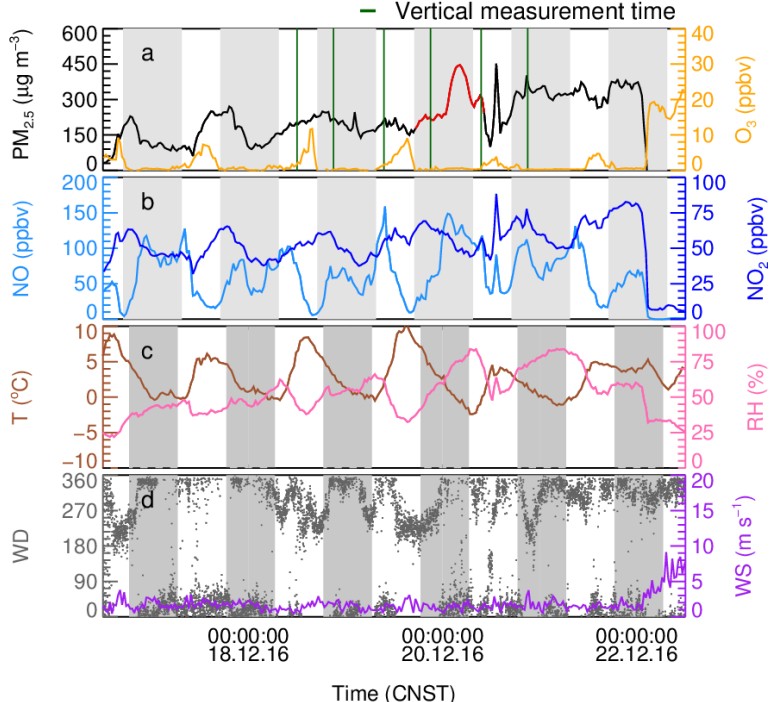

**Figure 2.** Time series of **(a)** PM$_{2.5}$ and O$_3$, **(b)** NO and NO$_2$, **(c)** temperature (T) and relative humidity (RH), **(d)** wind direction (WD) and wind speed (WS) from December 16 to 22, 2016 at PKU site in Beijing, China. The shaded region represents the nighttime periods. Red line in panel **(a)** shows an example of fast PM$_{2.5}$ enhancement on the night of December 19, and the green lines are the time periods when the vertical measurements conducted in IAP site.

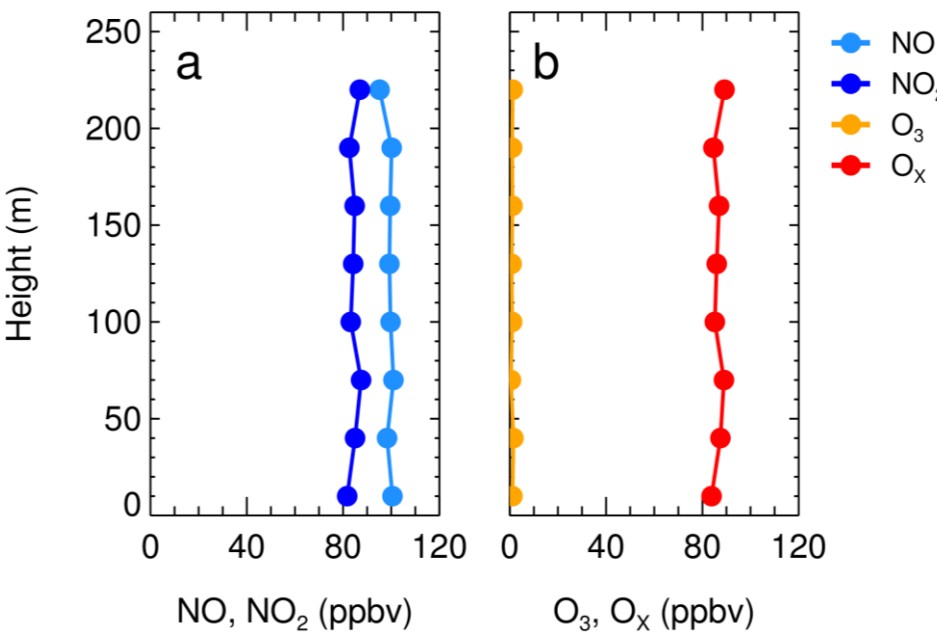


**Figure 3.** Vertical profiles of NO and $NO_2$ (**a**), $O_3$ and $O_x$ (**b**) at 20:38-21:06 on the
night of December 20, 2016.


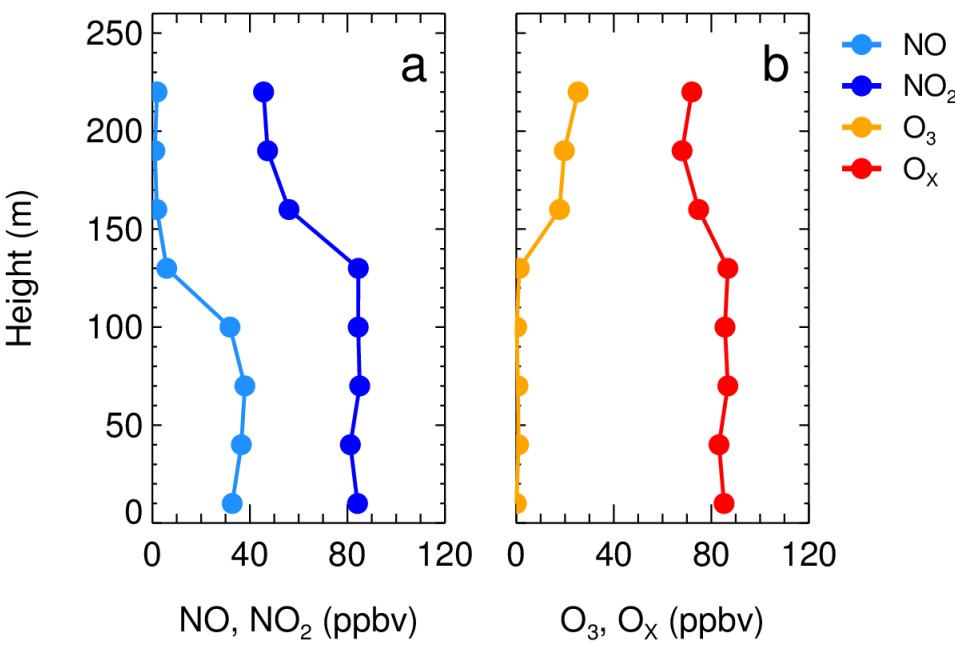


**Figure 4.** $O_x$ missing case presented by the vertical profiles of (**a**) NO and $NO_2$, (**b**)
$O_3$ and $O_x$ at 20:38-21:13 on the night of December 19, 2016.

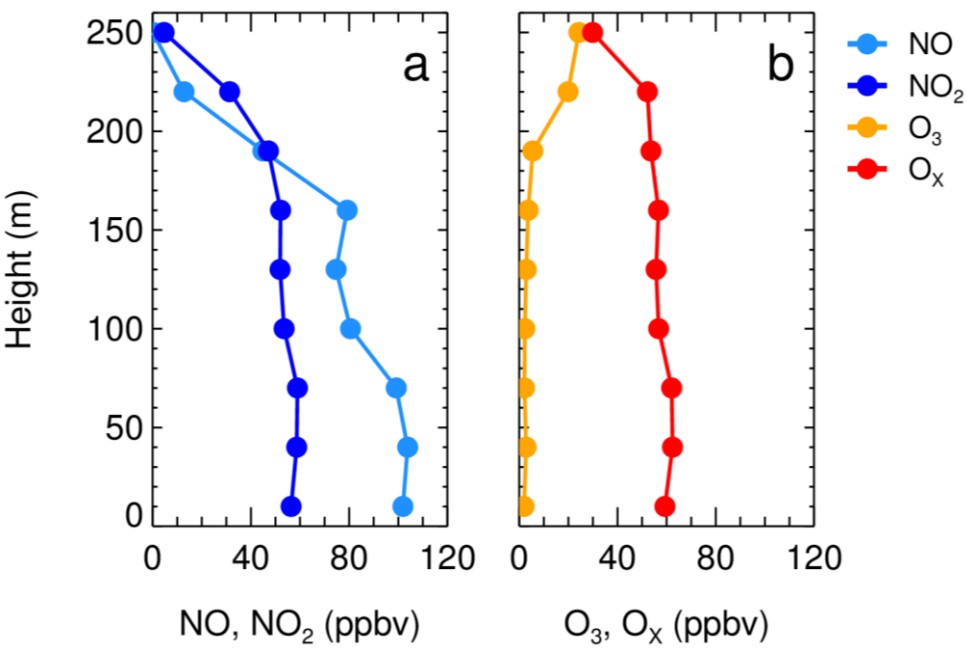


**Figure 5.** Vertical profiles of (**a**) NO and NO$_2$, (**b**) O$_3$ and O$_x$ at 09:06-09:34 in the
morning of December 20, 2016.


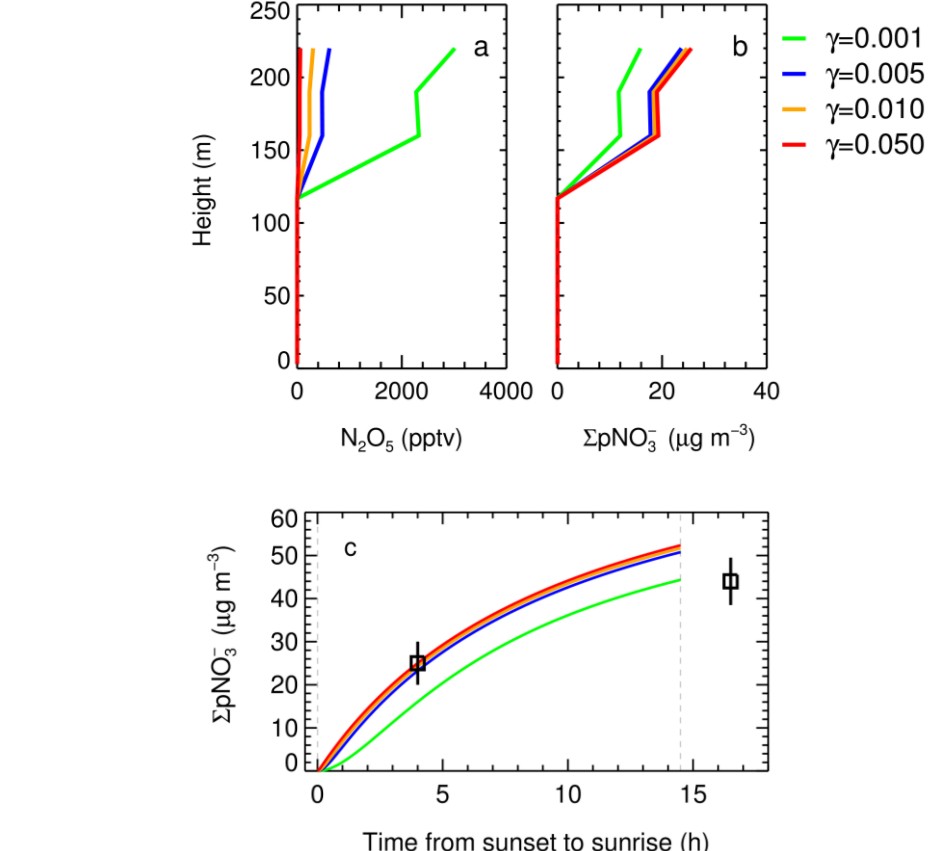


**Figure 6.** Base case ($\gamma$=0.005) and sensitivity tests of the vertical profile on the night of December 19 at different $N_2O_5$ uptake coefficients, including **(a)** the mixing ratio of $N_2O_5$ at 21:00, **(b)** the integral $pNO_3^-$ production from sunset to 21:00, **(c)** the time series of the integral $pNO_3^-$ formed at 240 m via $N_2O_5$ uptake from sunset (17:00) to sunrise (07:30, nighttime length = 14.5 h), the squares represents the $pNO_3^-$ equivalent weight from the observed $O_x$ missing in the two vertical measurements ~21:00 and ~09:30 in the following morning.

765

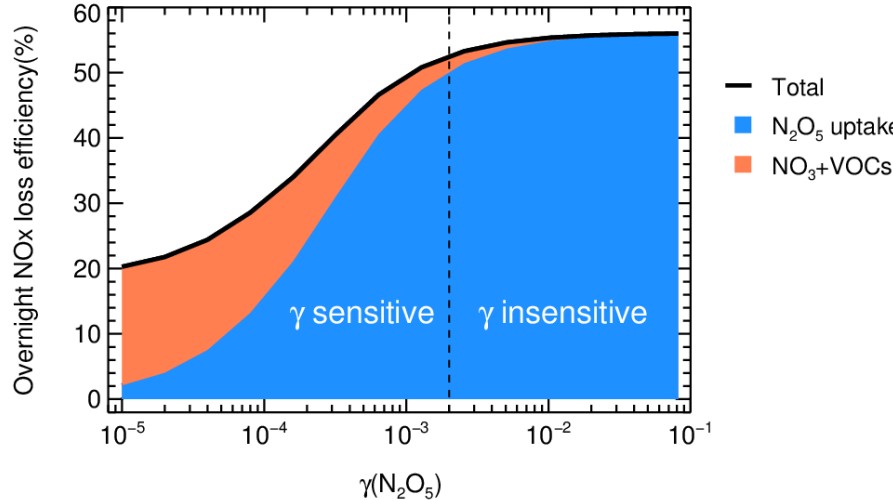

766

**Figure 7.** The dependence of overnight $NO_x$ loss via $N_2O_5$ uptake on $\gamma_{N2O5}$ in a typical winter pollution condition. The initial $NO_2$ and $O_3$ were set to 60 ppbv and 30 ppbv, respectively, $S_a$ was set to 3000 $\mu m^2$ $cm^{-3}$, the $ClNO_2$ yield was zero and $k_{NO3}$ was 0.02 $s^{-1}$. The reaction time was set to 14.5 h. The blue and orange zone represent the contribution by $NO_3$+VOCs and $N_2O_5$ uptake, the dashed line ($\gamma$ = 0.002, when $N_2O_5$ uptake contribute to 90% of the maximum $NO_x$ loss) divide the loss into $\gamma$ sensitive and insensitive region. The maximum nocturnal $NO_x$ loss by $NO_3$-$N_2O_5$ chemistry is 56%.

775

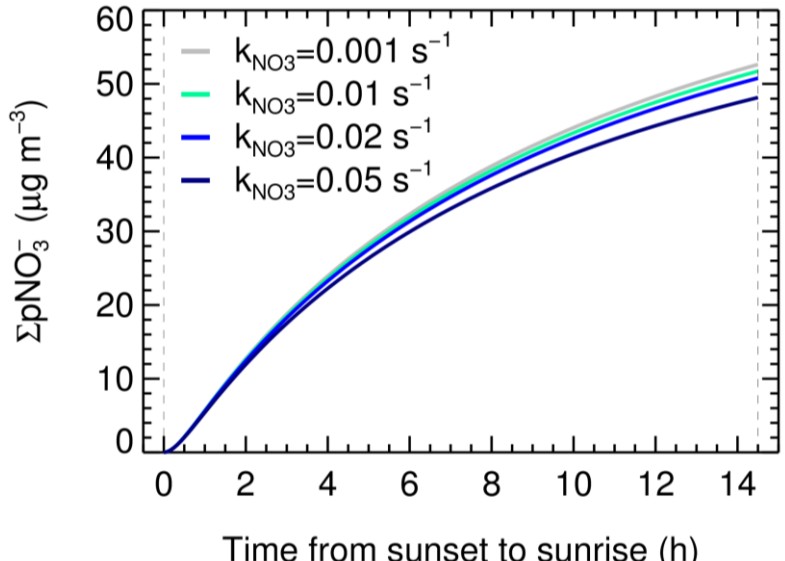

**Figure 8.** Base case ($k_{NO3}$=0.02 s$^{-1}$) and sensitivity tests of the integral pNO$_3^-$ formed at 240 m via N$_2$O$_5$ uptake at different NO$_3$ reactivity (0.001 s$^{-1}$, 0.01 s$^{-1}$, 0.05 s$^{-1}$) on the whole night of December 19, 2016.

**Table 1.** List of the parameter sets in base case and sensitivity tests.

| Cases | $k_{NO3}$ (s$^{-1}$) | $\gamma_{N2O5}$ |
|---|---|---|
| Base case | 0.02 | 0.005 |
| $k_{NO3}$ test 1 | 0.001 | 0.005 |
| $k_{NO3}$ test 2 | 0.01 | 0.005 |
| $k_{NO3}$ test 3 | 0.05 | 0.005 |
| $\gamma_{N2O5}$ test 1 | 0.02 | 0.001 |
| $\gamma_{N2O5}$ test 2 | 0.02 | 0.01 |
| $\gamma_{N2O5}$ test 3 | 0.02 | 0.05 |