# Peer review of "Fast particulate nitrate formation via N2O5 uptake aloft in winter Beijing"

_Atmospheric Chemistry and Physics, 2017_

## Referee Comment (RC1) · Anonymous Referee #1 · 19 Mar 2018

In this manuscript, Wang et al. estimated particulate nitrate formation from $N_2O_5$ uptake at air aloft (~100 m) over urban Beijing, based on the vertical measurements of ozone, NO, and $NO_2$ on one episode day obtained at a tower and box model calculations. With a number of assumptions, the model suggested significant particle nitrate formation of up to 50 μg m$^{-3}$ from the $N_2O_5$ heterogeneous hydrolysis in air masses aloft for this episode. The model calculations also suggested that the oxidation of $NO_x$ to nitrate was maximized once $N_2O_5$ uptake coefficient was over 0.0017, and became insensitive with higher uptake coefficient.

The topic of the heterogeneous process of $NO_x$ is of interest to the community, and the vertical measurements of chemical composition in winter haze are particularly valuable. My main concern, however, on the present work is that the model was poorly constrained by observations and had to reply on too many assumptions. The vertical measurements only included $O_3$, NO, and $NO_2$ which enabled calculation of $NO_3$ production, but several key parameters, such as $N_2O_5$, VOCs and aerosol surface area density, for loss of $NO_3$, $N_2O_5$ production and subsequent loss to nitrate were not measured, making it very difficult, if not impossible, to evaluate nighttime reactions of $NO_x$ and for nitrate formation. In addition, the analysis was only based on one profile measured in the early evening, and too many assumptions in the model calculations were not well justified. All these make it very difficult to judge the validity of the conclusions drawn from the analysis. The authors are advised to carefully consider and reduce these uncertainties which could lead to large bias and possible errors.

**Specific comments:**

L16. Please elaborate what the simultaneous measurements were conducted.

L22. Please define the potential of $pNO_3^-$.

L57-58. It is not clear how the $N_2O_5$ uptake coefficient in winter will be different from summer.

L96. the abbreviation of 'IAP' should be spelled out upon the first mention in the text.

L110. Please specify which instruments were installed on board a movable cabin.

L111-113. Describe in more detail about the two light-weight instruments for vertical measurements of $O_3$, NO and $NO_2$. How were they calibrated, was there any intercomparison with conventional monitors?

L124. Were the daily cycles conducted in the same time periods for every day?

L129-131. The assumptions of no NO influence, no physical mixing, and no transport of the air mass may not be valid here. Without continuous or intermittent measurement as constraints, it is difficult to know the evolution of the air masses, as such one cannot testify the validity of these assumptions.

L143-147, The low theoretical equilibrium ratio of $NO_3$ to $N_2O_5$ at the low-temperature condition may not necessarily mean that the $N_2O_5$ formation dominates the $NO_3$ loss. More evidence is needed here. The $k_{NO3}$ value of 0.02 s$^{-1}$ assumed in the present study was much higher than other

studies, for example, Brown et al., 2016, in which the $NO_3$ reaction with VOC contributes more than half of the total $NO_3$ loss.

L148-150. It is not clear how the author determined the initial concentration of $NO_2$ and $O_3$, which could affect the integrated concentrations. Were you using the iterative method suggested by Wagner et al., 2013? The retrieved results should be included in the supplementary. Any measurement constraints were used to validate this calculation?

L146. "than" should be "that".

L162. Please clarify the exact time period for the model running from sunset to sunrise.

L164-168. There are some flaws in the Eq.1 to 5 of the box model. To simplify the differential equation, the author assumes an equilibrium between $NO_3$ and $N_2O_5$ in Eq. 3 and 4, which means the loss rate of $NO_2$ through $R_2$ and production rate from $R_3$ should be equal. Therefore, the $NO_2$ loss rate will equal to the reaction rate of R1, which contradicts Eq. 2. As suggested by Wagner et al., 2013, "the assumption of equilibrium leads to an error which accumulates as the equations are integrated". This could affect the results when retrieving the initial $NO_2$ concentration and subsequent model simulation. To be more accurate, I suggest the author use the explicit equations suggested by Wagner et al., 2013.

L169-L179. The surface area was calculated based on the measurement of particle size distribution from 0.01 to 0.6 μm and could be underestimated due to the lack of information of larger particles, resulting in large uncertainty in the calculated uptake coefficients. It is necessary to provide an uncertainty estimation of how much this will affect the results.

L240-241, L245-247, L257-258. The author attributed the lower $O_x$ level at high altitude on the night of December 19 to missing sinks of $O_x$ with high $N_2O_5$ uptake, but it could also be a result of the continuous emissions of NOx near the ground leading to accumulation of $NO_2$ within the nocturnal boundary layer with a height around 100m. So the $O_x$ level in the residual layer and surface doesn't have to be conserved. Here it would be good to show vertical information on meteorological parameters.

L264-265, Wrong figure number referenced here.

L265. Please explicitly define the equation used for calculating the nitrate accumulation.

L262-267. This model simulation assumes an ideal condition with no NO concentration above 150m from the sunset to 21:00, which cannot be substantiated. Thus the calculated accumulation of nitrate is questionable.

L281-282. Please elaborate how to deal with calculations of mixing.

L282-285. If the author's hypothesis is true, it should be able to observe a sharp increase of particulate nitrate at the ground site in the morning of December 20. Any evidence on that?

L309. Eq.6 is incorrect. The particle nitrate formation is twice of $N_2O_5$ loss if assuming $ClNO_2$ yield is zero.

L328, "coral" should be "coal".

L342-343. A possible reason for the small difference on $k_{NO3}$ variation could be the $NO_3$ change (via VOC loss) were unaccounted for in Eq.2 that used to retrieve the initial concentration. Comparison between the full differential equations and simplified calculation is required to validate the results.

L351, delete "to be zero".

L357, add "that" after "found".

---

## Referee Comment (RC2) · Anonymous Referee #2 · 20 Mar 2018

**"Large particulate nitrate formation from $N_2O_5$ uptake in a chemically reactive layer aloft during wintertime in Beijing" by Wang et al.**

The authors explore the mechanisms for particulate nitrate ($pNO_3^-$) during wintertime haze events in Beijing, China. Comparing simultaneous ground-based and tower-based observations, the authors investigated the significance of $pNO_3^-$ via N2O5 heterogeneous uptake as a function of altitude. The work shows the effects of the $pNO_3^-$ formed aloft on the surface $PM_{2.5}$ the following the morning. Given the significance of this work, I recommend this manuscript for publication after significant revisions.

1. Although the experiment design is well thought and the analysis appears to be solid, the technical writing needs significant improvement. I recommend the authors to use professional technical writing services in English to improve the penmanship and eliminate any grammatical errors. Example sentences to be reviewed carefully and reformulated are line 66 – 70, line 178-179, 180 – 183, 186 – 190, 194-195, 205-213, 242 – 244, 275-278, 292-295 etc.

2. I am assuming eq.1 (line 164) is for the nitrate radical production rate ($P_{NO3}$), not the rate of change in $O_3$. As the authors mentioned the availability of $O_3$ is driven by its reaction with NO.

3. Use subscript for $O_x$ throughout the text

4. The authors define and discuss "particle nitrate convert efficiency" (sigma) in line 305-310. Chang et al.[1] gives an excellent review of N2O5 chemistry and I suggest the authors read this as they discuss and introduce parameters regarding N2O5 conversion. I do not believe it is necessary to introduce a new parameter "particle nitrate convert efficiency" in this case.

5. In that regard, the authors need to extend the literature search and include more references on N2O5 heterogeneous uptake and wintertime haze events outside the Beijing area. For more references on relevant topic, review publications of Chang et al.[1-2], Lurmann et al. [3], Brown et al. [4], Green et al. [5], Wang et al. [6], Prabhakar et al.[7] etc.

1. Chang, W.; Bhave, P.; Brown, S.; Riemer, N.; Stutz, J.; Dabdub, D., Heterogeneous atmospheric chemistry, ambient measurements, and model calculations of $N_2O_5$: A review. *Aerosol Sci. Technol.* **2011,** *45* (6), 665-695. DOI 10.1080/02786826.2010.551672.
2. Chang, W. L.; Brown, S. S.; Stutz, J.; Middlebrook, A. M.; Bahreini, R.; Wagner, N. L.; Dubé, W. P.; Pollack, I. B.; Ryerson, T. B.; Riemer, N., Evaluating $N_2O_5$ heterogeneous hydrolysis parameterizations for CalNex 2010. *J. Geophys. Res.: Atmos.* **2016,** *121* (9), 5051-5070. DOI 10.1002/2015JD024737.
3. Lurmann, F. W.; Brown, S. G.; McCarthy, M. C.; Roberts, P. T., Processes influencing secondary aerosol formation in the San Joaquin Valley during winter. *J. Air Waste Manage. Assoc.* **2006,** *56* (12), 1679-1693. DOI 10.1080/10473289.2006.10464573.

4.      Brown, S. G.; Roberts, P. T.; McCarthy, M. C.; Lurmann, F. W.; Hyslop, N. P., Wintertime vertical variations in particulate matter (PM) and precursor concentrations in the San Joaquin Valley during the California Regional Coarse PM/Fine PM Air Quality Study. *J. Air Waste Manage. Assoc.* **2006,** *56* (9), 1267-1277. DOI 10.1080/10473289.2006.10464583.

5.      Green, M. C.; Chow, J. C.; Watson, J. G.; Dick, K.; Inouye, D., Effects of snow cover and atmospheric stability on winter $PM_{2.5}$ concentrations in Western U.S. valleys. *J. Appl. Meteor. Climatol.* **2015,** *54* (6), 1191-1201. DOI 10.1175/JAMC-D-14-0191.1.

6.      Wang, G.; Zhang, R.; Gomez, M. E.; Yang, L.; Levy Zamora, M.; Hu, M.; Lin, Y.; Peng, J.; Guo, S.; Meng, J.; Li, J.; Cheng, C.; Hu, T.; Ren, Y.; Wang, Y.; Gao, J.; Cao, J.; An, Z.; Zhou, W.; Li, G.; Wang, J.; Tian, P.; Marrero-Ortiz, W.; Secrest, J.; Du, Z.; Zheng, J.; Shang, D.; Zeng, L.; Shao, M.; Wang, W.; Huang, Y.; Wang, Y.; Zhu, Y.; Li, Y.; Hu, J.; Pan, B.; Cai, L.; Cheng, Y.; Ji, Y.; Zhang, F.; Rosenfeld, D.; Liss, P. S.; Duce, R. A.; Kolb, C. E.; Molina, M. J., Persistent sulfate formation from London Fog to Chinese haze. *Proceedings of the National Academy of Sciences* **2016,** *113* (48), 13630. DOI.

7.   Prabhakar, G., C. Parworth, X. Zhang, H. Kim, D. Young, A.J. Beyersdorf, L.D. Ziemba, J.B. Nowak, T.H. Bertram, I.C. Faloona, Q. Zhang, and C.D. Cappa, *Observational assessment of the role of nocturnal residual-layer chemistry in determining daytime surface particulate nitrate concentrations.* Atmos. Chem. Phys. Discuss., 2017. **2017**: p. 1-58

---

## Author Response (AR1)

**Response to Reviewers**

We thank the reviewers for their careful reading and their constructive comments on our manuscript. As detailed below, the reviewer's comments are shown as italicized font, our response to the comments are normal font. New or modified text is in blue. All of the line numbers refer to Manuscript ID: acp-2017-1217.

**Reviewer: #1**

*The topic of the heterogeneous process of NOx is of interest to the community, and the vertical measurements of chemical composition in winter haze are particularly valuable. My main concern, however, on the present work is that the model was poorly constrained by observations and had to reply on too many assumptions. The vertical measurements only included $O_3$, NO, and $NO_2$ which enabled calculation of $NO_3$ production, but several key parameters, such as $N_2O_5$, VOCs and aerosol surface area density, for loss of $NO_3$, $N_2O_5$ production and subsequent loss to nitrate were not measured, making it very difficult, if not impossible, to evaluate nighttime reactions of NOx and for nitrate formation. In addition, the analysis was only based on one profile measured in the early evening, and too many assumptions in the model calculations were not well justified. All these make it very difficult to judge the validity of the conclusions drawn from the analysis. The authors are advised to carefully consider and reduce these uncertainties which could lead to large bias and possible errors.*

*1). L16. Please elaborate what the simultaneous measurements were conducted.*

Changed as following: "Simultaneous ground-based and tower-based measurements of $NO_x$ and $O_3$ were conducted…"

*2). L22. Please define the potential of $pNO_3^-$ .*

This sentence change to: "The nighttime integrated production of $pNO_3^-$ for …"

Here we deleted "potential".

*3). L57-58. It is not clear how the $N_2O_5$ uptake coefficient in winter will be different from summer.*

Since the properties of the aerosol particles (e.g., organic compounds, particle nitrate, liquid water contents, solubility, viscosity, etc.) and meteorological conditions and (e.g. temperature, relative humidity etc.) are different in summer and winter, these differences will led to changes of the $N_2O_5$ uptake coefficient. The explanation was added in Line 57-58: "This is because the properties of aerosol particles (e.g., organic compounds, particulate nitrate, liquid water contents, solubility, and viscosity) and meteorological conditions (e.g., temperature and relative humidity) differ between summer and winter (Chen et al., 2015; Zhang et al., 2007)."

**4). L96. The abbreviation of 'IAP' should be spelled out upon the first mention in the text.**

Changed accordingly.

**5). L110. Please specify which instruments were installed on board a movable cabin.**

Specified in the Line 110: "$NO_x$ and $O_3$ instruments were installed on board a movable cabin on the tower…"

**6). L111-113. Describe in more detail about the two light-weight instruments for vertical measurements of $O_3$, NO and $NO_2$. How were they calibrated, was there any intercomparison with conventional monitors?**

The detailed description after Line 113 was added: "$NO_x$ calibration was performed in the lab using a gas calibrator (TE-146i, Thermo Electron, USA) associated with a NO standard (9.8 ppmv). The $O_3$ calibration was done with an $O_3$ calibrator (TE 49i-PS), which was traceable to NIST (National Institute of Standards and Technology) standards annually. Before the campaign, the $NO_x$ monitor was compared with a Cavity Attenuated Phase Shift (CAPs) Particle Light Extinction Monitor, and the $O_3$ monitor was compared to a commercial $O_3$ analyzer (TE-49i, Thermo Electron, USA). Good agreement was found between the portable instruments and the conventional monitors."

**7). L124. Were the daily cycles conducted in the same time periods for every day?**

Yes, we conducted the daily cycle measurement in the similar time periods for the three days.

**8). L129-131. The assumptions of no NO influence, no physical mixing, and no transport of the air mass may not be valid here. Without continuous or intermittent measurement as constraints, it is difficult to know the evolution of the air masses, as such one cannot testify the validity of these assumptions.**

Since high $O_3$ concentrations (20 ppbv) at the high altitude (>150 m) were observed at night, the NO concentrations have to be zero for this kind of condition. We were using a box model for the interpretation of the observed dataset. In the framework of box model analysis, the assumption is the analyzed air mass were well mixed. The assumption of well mixing is plausible since the influence of physical mixing on the reaction rate of NO + $O_3$ could be neglected.

**9). L143-147. *The low theoretical equilibrium ratio of $NO_3$ to $N_2O_5$ at the low-temperature condition may not necessarily mean that the $N_2O_5$ formation dominates the $NO_3$ loss. More evidence is needed here. The $kNO_3$ value of 0.02 $s^{-1}$ assumed in the present study was much higher than other studies, for example, Brown et al., 2016, in which the $NO_3$ reaction with VOC contributes more than half of the total $NO_3$ loss.***

Yes, high $N_2O_5/NO_3$ is not means $N_2O_5$ heterogeneous uptake dominate the $NO_3$ loss. While during the polluted period in winter Beijing ($NO_2$ = 45 ppbv, Temperature = 273 K, Sa = 3000 $\mu m^2$ $cm^{-3}$), the pseudo first order loss rate of $N_2O_5$ heterogeneous uptake will be $1\times10^{-3}$ $s^{-1}$ , corresponding to the $N_2O_5$ uptake coefficient of $5\times10^{-3}$, and contributed to $NO_3$ loss rate of 0.4 $s^{-1}$, which is much higher than the direct $NO_3$ loss by the reaction of $NO_3$ with VOCs, even the $k_{NO3}$ set to a high value of 0.02 $s^{-1}$. Therefore we believe the $N_2O_5$ formation dominates the $NO_3$ loss in this study. With respect to $k_{NO3}$, Brown et al., (2016) shows the wintertime average $k_{NO3}$ in Hong Kong was about $6\times10^{-3}$ $s^{-1}$, and dominated by monoterpenes. Previous work showed the average $kNO_3$ is about 0.011 $s^{-1}$ in rural Beijing in summertime, and BVOCs is the dominating part (Wang et al., 2017; Wang et al., 2018). During wintertime, the BVOCs emission would decrease due to lower temperature and weak solar radiation, the $k_{NO3}$ set to 0.02 $s^{-1}$ in this study represents an upper value to some extent. The differences of this study with the campaign conducted in Hong Kong (Brown et al., 2016) may cause by the higher temperature and much lower aerosol surface area in Hong Kong (Temperature = 285 K, Sa $\approx$ 200 $\mu m^2$ $cm^{-3}$).

In Line 145, we rewrite the part as following: "During the polluted period in winter Beijing (here $NO_2$ = 45 ppbv, Temperature = 273 K, Sa = 3000 $\mu m^2$ $cm^{-3}$), the ratio of $N_2O_5$ to $NO_3$ is large enough, i.e., 450, the pseudo first order loss rate of $N_2O_5$ heterogeneous uptake will be $1\times10^{-3}$ $s^{-1}$ with the $N_2O_5$ uptake coefficient of $5\times10^{-3}$. $N_2O_5$ uptake would contribute an $NO_3$ loss rate of 0.4 $s^{-1}$, and much higher than the direct $NO_3$ loss by the reaction of $NO_3$ with VOCs, even the $k_{NO3}$ set to a high value of 0.02 $s^{-1}$. Therefore, $N_2O_5$ uptake was proposed to be dominantly responsible for the $NO_3$ loss and the initial s(t) was set to 1"

**10). L148-150. *It is not clear how the author determined the initial concentration of $NO_2$ and $O_3$, which could affect the integrated concentrations. Were you using the***

*iterative method suggested by Wagner et al., 2013? The retrieved results should be included in the supplementary. Any measurement constraints were used to validate this calculation?*

Yes, we used the iterative method suggested by Wagner et al., 2013. The initial $NO_2$ and $O_3$ concentration were derived according to Eq. 1 and Eq. 2, $O_3$ and $NO_2$ are integrated backward in time to sunset. The average initial $NO_2$ and $O_3$ above 150 m at sunset time is about $61 \pm 3$ ppbv and $27 \pm 6$ ppbv, respectively. The measured $NO_2$ concentration in PKU site at sunset time (local time, 16:55) is 61 ppbv and show good consistent with the model result.

Changed as following in Line 299: "The average initial $NO_2$ and $O_3$ above 150 m at sunset time is about $61 \pm 3$ ppbv and $27 \pm 6$ ppbv, respectively. The measured $NO_2$ concentration in PKU site at sunset time (local time, 16:55) is 61 ppbv and show good consistent with the model result."

**11). L146. "than" should be "that".**

Corrected accordingly.

**12). L162. Please clarify the exact time period for the model running from sunset to sunrise.**

Here we clarified as: "Sunset and sunrise time during the measurement is 16:55 and 07:30 (Chinese National Standard Time, CNST) and the length of night was about 14.5 h, the model is run from sunset to sunrise with the running time set to 14.5 h."

**13). L164-168. There are some flaws in the Eq.1 to 5 of the box model. To simplify the differential equation, the author assumes an equilibrium between $NO_3$ and $N_2O_5$ in Eq. 3 and 4, which means the loss rate of $NO_2$ through R2 and production rate from R3 should be equal. Therefore, the $NO_2$ loss rate will equal to the reaction rate of R1, which contradicts Eq. 2. As suggested by Wagner et al., 2013, "the assumption of equilibrium leads to an error which accumulates as the equations are integrated". This could affect the results when retrieving the initial $NO_2$ concentration and subsequent model simulation. To be more accurate, I suggest the author use the explicit equations suggested by Wagner et al., 2013.**

Thanks for the suggestion, we corrected these equations accordingly, and the fixed $k_{NO3}$ and $k_{N2O5}$ first, set s(t) to 1 in the first step and then iterate the s(t) till the difference between two iteration less than 0.005. The explicit equations changed as:

"(Eq. 1)  $\frac{d\,[O_3]}{dt} = -k_{NO2+O3}[O_3][NO_2]$

(Eq. 2)   $\dfrac{d\,[NO_2]}{dt} = -(1 + s(t)) \times k_{NO2+O3}[O_3][NO_2]$

(Eq. 3)   $\dfrac{d\,[NO_3 + N_2O_5]}{dt} = k_{NO2+O3}[O_3][NO_2] - k_{N2O5}[N_2O_5] - k_{NO3}[NO_3]$

(Eq. 4)   $\dfrac{[N_2O_5]}{[NO_3]} = k_{eq}[NO_2]$

(Eq. 5)   $s(t) = \dfrac{\int_0^t k_{N2O5} \cdot [N_2O_5] dt + [N_2O_5]_t}{[O_3](0) - [O_3](t)}$

(Eq. 6)   $k_{N2O5} = \dfrac{C \times S_a \times \gamma_{N2O5}}{4}$  ,,

**14). L169-L179. The surface area was calculated based on the measurement of particle size distribution from 0.01 to 0.6 μm and could be underestimated due to the lack of information of larger particles, resulting in large uncertainty in the calculated uptake coefficients. It is necessary to provide an uncertainty estimation of how much this will affect the results.**

During the study period, the particle number and size distribution (PNSD) larger than 0.7 μm is unavailable, it is different to quantify the contribution from lager particles. While during the following polluted episode (2017-01-01 to 2017-01-07), PNSD of $PM_{2.5}$ data are available, we found particle smaller than 0.7 μm dominated more than 95% aerosol surface area, the similar result also represented in Germany and summer Beijing (Crowley et al., 2010; Wang et al., 2018). The underestimation of $S_a$ in this study (5%) could lead to the overestimation of $N_2O_5$ 3.6% - 4.2%, and underestimation of $pNO_3^-$ with 0.2% - 2.5% considering the $N_2O_5$ uptake coefficient varied from $1 \times 10^{-3}$ to 0.05.

We added the description in Line 179: "Although the PNSD information for particles larger than 0.7 μm was not valid during the study period, the particles smaller than 0.7 μm dominated more than 95% of the aerosol surface area in a subsequent pollution episode (01/01/2017 to 01/07/2017), and similar results also were reported in other studies (e.g., Crowley et al., 2010a; Wang et al., 2018). The possible lower bias of $S_a$ (5%) only led to a small overestimation of $N_2O_5$, i.e., 3.6%–4.2%, and an underestimation of $pNO_3^-$ of 0.2%–2.5% when $\gamma_{N2O5}$ varied from $1 \times 10^{-3}$ to 0.05."

**15). L240-241, L245-247, L257-258. The author attributed the lower Ox level at high altitude on the night of December 19 to missing sinks of Ox with high $N_2O_5$ uptake, but it could also be a result of the continuous emissions of NOx near the ground leading to accumulation of NO2 within the nocturnal boundary layer with a height around 100m. So the Ox level in the residual layer and surface doesn't have to be conserved. Here it would be good to show vertical information on meteorological parameters.**

The emission of NO would not influence the sum of Ox (=$O_3$+$NO_2$). The emission of $NO_2$ would led to higher $O_x$ at the surface layer. Nevertheless, the vertical profile measurement showed no vertical gradient of $NO_2$ lower than 150 m so that we do not think there could be a significant emission of $NO_2$. And therefore, the $O_x$ level shall be conserved between the nocturnal boundary layer and the residual layer since no $O_3$ is produced at night. In addition, the nocturnal boundary layer is determined to be about 340 m through the vertical profile of temperature during the same period in Beijing (Zhong et al., 2017).

*16). L264-265, Wrong figure number referenced here.*

Corrected accordingly.

*17). L265. Please explicitly define the equation used for calculating the nitrate accumulation.*

We listed the calculation equation as following:

"$$\sum \text{pNO}_3^- = \int_0^t (2-f) \cdot k_{\text{N2O5}} \cdot [\text{N}_2\text{O}_5] dt \qquad (7)$$"

*18). L262-267. This model simulation assumes an ideal condition with no NO concentration above 150 m from the sunset to 21:00, which cannot be substantiated. Thus the calculated accumulation of nitrate is questionable.*

Due to the strong thermal inversion during winter haze episode (e.g. Zhong et al., 2017), the isolation is existed more easily in vertical scale in urban Beijing, the air mass in upper layer is not easily affected by surface NO emission. The theoretical framework of the box model we used is same as Wagner et al., (2013) and Yun et al., (2018). The model allow us to accumulate the $pNO_3^-$ till sunrise, which shows an upper limit of the nitrate production via $N_2O_5$ uptake in the upper layer. In addition, at sunset time, we observed significant $O_3$ presented at the near surface layer. Before $O_3$ is fully titrated away, the NO concentrations shall be zero for the sunset time.

*19). L281-282. Please elaborate how to deal with calculations of mixing.*

We rewrite the vertical mixing in L279-283 as following: "Zhong et al. (2017) showed that the NBL and PBL both were at 340 m from December 19 to 20, 2016 in Beijing. Daytime vertical downward transportation was helpful in mixing the air mass within the PBL. Assuming the newly formed $pNO_3^-$ aloft from 150 m to 340 m is 50 µg m$^{-3}$ during the nighttime and well mixed within the PBL by the next morning, the enhancement to the surface layer ($\Delta pNO_3^-$) can be simplified to the calculation in

Eq. 8 as following:

$$\Delta pNO_3 = \frac{\int_0^{150} P(pNO_3)dH + \int_{150}^{340} P(pNO_3)dH}{340} \tag{8}$$

Here, $P(pNO_3^-)$ is the integral production of $pNO_3^-$ and $H$ represents height. Owing to high NO below 150 m, the $pNO_3^-$ formation via $N_2O_5$ uptake was zero. The enhancement of $pNO_3^-$ from 150 m to 340 m was calculated as 28 µg m$^{-3}$,"

**20). L282-285. If the author's hypothesis is true, it should be able to observe a sharp increase of particulate nitrate at the ground site in the morning of December 20. Any evidence on that?**

The particulate nitrate measurement is not available in this study, but as labelled in Figure 2(a), the red line showed PM concentration had a sharp increase of ~60 µg m$^{-3}$, which was purposed to be consist with the result considering a large proportion of particulate nitrate in PM mass concentration, especially during winter polluted episode in Beijing (e.g., Zheng et al., 2015).

**21). L309. Eq.6 is incorrect. The particle nitrate formation is twice of $N_2O_5$ loss if assuming $ClNO_2$ yield is zero.**

Thanks, we corrected accordingly.

**22). L328, "coral" should be "coal".**

Corrected accordingly.

**23). L342-343. A possible reason for the small difference on $kNO_3$ variation could be the $NO_3$ change (via VOC loss) were unaccounted for in Eq.2 that used to retrieve the initial concentration. Comparison between the full differential equations and simplified calculation is required to validate the results.**

The full differential equations was used to recalculate the $pNO_3^-$ variation on $k_{NO3}$. The s(t) decreased from 1 to 0.99 even $k_{NO3}$ set to 0.05, and the difference between the full differential equations and simplified calculation is negligible, suggested that the calculation result is valid.

**24). L351, delete "to be zero".**

Corrected accordingly.

**25). L357, add "that" after "found".**

Corrected accordingly.

**Reviewer: #2**

*The authors explore the mechanisms for particulate nitrate (pNO₃⁻ ) during wintertime haze events in Beijing, China. Comparing simultaneous ground-based and tower-based observations, the authors investigated the significance of $pNO_3^-$ via $N_2O_5$ heterogeneous uptake as a function of altitude. The work shows the effects of the $pNO_3^-$ formed aloft on the surface $PM_{2.5}$ the following the morning. Given the significance of this work, I recommend this manuscript for publication after significant revisions.*

We thank for the Reviewer #2's constructive comments and suggestions to improve the quality of our manuscript.

**1). although the experiment design is well thought and the analysis appears to be solid, the technical writing needs significant improvement. I recommend the authors to use professional technical writing services in English to improve the penmanship and eliminate any grammatical errors. Example sentences to be reviewed carefully and reformulated are line 66-70, line 178-179, 180-183, 186-190, 194-195, 205-213, 242-244, 275-278, 292-295 etc.**

The resubmitted manuscript has been edited by a professional service in English.

**2). I am assuming eq.1 (line 164) is for the nitrate radical production rate ($PNO_3$), not the rate of change in $O_3$. As the authors mentioned the availability of $O_3$ is driven by its reaction with NO.**

Yes, Eq. 1 is the production of nitrate radical, but $O_3$ is also one reactant of this reaction. As the production of $NO_3$ takes place, the $O_3$ is consumed. This reaction is more important for $O_3$ losses for the conditions of the high-altitude (>150 m) air masses of which the reaction pathway of $O_3$ + NO is negligible due to the presence of zero NO.

**3). Use subscript for Ox throughout the text**

Corrected accordingly.

*4). the authors define and discuss "particle nitrate convert efficiency" (sigma) in line 305- 310. Chang et al. 1 gives an excellent review of $N_2O_5$ chemistry and I suggest the authors read this as they discuss and introduce parameters regarding $N_2O_5$ conversion. I do not believe it is necessary to introduce a new parameter "particle nitrate convert efficiency" in this case.*

Thanks for your suggestion, Chang et al. (2011) reviewed the $N_2O_5$ chemistry systematically and comprehensively. With respect to $N_2O_5$ conversion, Chang et al., focused on the contribution to overnight $NO_x$ loss. Here we revised the parameter to "Overnight $NO_x$ loss efficiency ($\varepsilon$)", which also indicates the nitrate formation capacity. The equation changed as following:

(Eq. 9)
$$\varepsilon = \frac{\int_0^t 2 \times k_{N2O5} \cdot [N_2O_5]dt + \int_0^t k_{NO3} \cdot [NO_3]dt}{[NO_2](0)}$$

Here the consumed $NO_3$ with VOCs and $N_2O_5$ uptake regarded as the effective $NO_x$ loss. The Figure 7 changed the Y-axis and we did not normalize the loss efficiency, which shows the similar result with previous figure version."

[Figure]

**Figure 7.** The dependence of overnight $NO_x$ loss on $N_2O_5$ uptake on $\gamma_{N2O5}$ in a typical winter pollution condition. The initial $NO_2$ and $O_3$ set to 60 ppbv and 30 ppbv, respectively, $S_a$ set to 3000 $\mu m^2$ $cm^{-3}$, the $ClNO_2$ yield is zero and $k_{NO3}$ is 0.02 $s^{-1}$. The reaction time set to 14.5 h. The blue and orange zone represent the contribution of $NO_3$+VOCs and $N_2O_5$ uptake, the dashed line ($\gamma$ = 0.002, when $N_2O_5$ uptake contribute to 90% of the maximum $NO_x$ loss) divide the loss into $\gamma$ sensitive and insensitive region. The maximum nocturnal $NO_x$ loss by $NO_3$-$N_2O_5$ chemistry is 56%.

*5). In that regard, the authors need to extend the literature search and include more references on $N_2O_5$ heterogeneous uptake and wintertime haze events outside the Beijing area. For more references on relevant topic, review publications of Chang et*

*al. 1-2 , Lurmann et al. 3 , Brown et al. 4 , Green et al. 5 , Wang et al. 6 , Prabhakar et al. 7 etc.*

*1. Chang, W.; Bhave, P.; Brown, S.; Riemer, N.; Stutz, J.; Dabdub, D., Heterogeneous atmospheric chemistry, ambient measurements, and model calculations of $N_2O_5$: A review. Aerosol Sci. Technol. **2011,** 45 (6), 665-695. DOI 10.1080/02786826.2010.551672.*

*2. Chang, W. L.; Brown, S. S.; Stutz, J.; Middlebrook, A. M.; Bahreini, R.; Wagner, N. L.; Dubé, W. P.; Pollack, I. B.; Ryerson, T. B.; Riemer, N., Evaluating $N_2O_5$ heterogeneous hydrolysis parameterizations for CalNex 2010. J. Geophys. Res.: Atmos. **2016,** 121 (9), 5051-5070. DOI 10.1002/2015JD024737.*

*3. Lurmann, F. W.; Brown, S. G.; McCarthy, M. C.; Roberts, P. T., Processes influencing secondary aerosol formation in the San Joaquin Valley during winter. J. Air Waste Manage. Assoc. **2006,** 56 (12), 1679-1693. DOI 10.1080/10473289.2006.10464573.*

*4. Brown, S. G.; Roberts, P. T.; McCarthy, M. C.; Lurmann, F. W.; Hyslop, N. P., Wintertime vertical variations in particulate matter (PM) and precursor concentrations in the San Joaquin Valley during the California Regional Coarse PM/Fine PM Air Quality Study. J. Air Waste Manage. Assoc. **2006,** 56 (9), 1267-1277. DOI 10.1080/10473289.2006.10464583.*

*5. Green, M. C.; Chow, J. C.; Watson, J. G.; Dick, K.; Inouye, D., Effects of snow cover and atmospheric stability on winter $PM_{2.5}$ concentrations in Western U.S. valleys. J. Appl. Meteor. Climatol. **2015,** 54 (6), 1191-1201. DOI 10.1175/JAMC-D-14-0191.1.*

*6. Wang, G.; Zhang, R.; Gomez, M. E.; Yang, L.; Levy Zamora, M.; Hu, M.; Lin, Y.; Peng, J.; Guo, S.; Meng, J.; Li, J.; Cheng, C.; Hu, T.; Ren, Y.; Wang, Y.; Gao, J.; Cao, J.; An, Z.; Zhou, W.; Li, G.; Wang, J.; Tian, P.; Marrero-Ortiz, W.; Secrest, J.; Du, Z.; Zheng, J.; Shang, D.; Zeng, L.; Shao, M.; Wang, W.; Huang, Y.; Wang, Y.; Zhu, Y.; Li, Y.; Hu, J.; Pan, B.; Cai, L.; Cheng, Y.; Ji, Y.; Zhang, F.; Rosenfeld, D.; Liss, P. S.; Duce, R. A.; Kolb, C. E.; Molina, M. J., Persistent sulfate formation from London Fog to Chinese haze. Proceedings of the National Academy of Sciences **2016,** 113 (48), 13630. DOI.*

*7. Prabhakar, G., C. Parworth, X. Zhang, H. Kim, D. Young, A.J. Beyersdorf, L.D. Ziemba, J.B. Nowak, T.H. Bertram, I.C. Faloona, Q. Zhang, and C.D. Cappa, Observational assessment of the role of nocturnal residual-layer chemistry in determining daytime surface particulate nitrate concentrations. Atmos. Chem. Phys. Discuss., 2017. **2017**: p. 1-58*

Thanks for the suggestion and we compared our results with these references that concern the winter haze event in other region, and cited these work in the revised manuscript.

**References**

[revised manuscript text omitted]

$$\text{(R4)} \quad NO_3 + VOCs \rightarrow Products \hspace{3cm} \text{(R4)}$$

$$\text{(R5)} \quad N_2O_5 + (H_2O \text{ or } Cl^-) \rightarrow (2\text{-}f)\, NO_3^- + f\, ClNO_2 \hspace{1cm} \text{(R5)}$$

Following the work of Wagner et al. (2013), the box model can be solved  using six equations (Eqs. 1-6). In the framework, $O_3$ is only lost via the reaction of $NO_2$ $+$ $O_3$ and the change in the $O_3$ concentration can  be expressed as Eq. 1. Eq. 2 can express the losses of $NO_2$. Here, the s(t) is between 0 and 1 and expressed as Eq. 5. The s(t) favors 0 when direct loss of $NO_3$ dominates and favors 1 when $N_2O_5$   uptake dominates $NO_3$ loss . The model calculation had two steps. The first step was to calculate the mixing ratio of $NO_2$ and $O_3$ at time zero (herein designated as sunset). According to Eqs. 1 and 2, the initial $NO_2$ (t=0) and $O_3$ (t=0) concentrations can then be integrated backward in time starting with the measured concentrations of $NO_2$ and $O_3$ at each height. During the pollution period in winter in Beijing ($NO_2$ = 45 ppbv, Temperature = 273 K, $S_a$ = 3000 μm² cm⁻³), the ratio of $N_2O_5$ to $NO_3$ is large enough, i.e., 450. The pseudo-first-order loss rate of $N_2O_5$ heterogeneous uptake will be $1\times10^{-3}$ s⁻¹, with a $N_2O_5$ uptake coefficient of $5\times10^{-3}$. $N_2O_5$ uptake would contribute an $NO_3$ loss rate of 0.4 s⁻¹, which is much higher than the direct $NO_3$ loss through the reaction of $NO_3$ with VOCs, even with the $k_{NO3}$ set to a high value of 0.02 s⁻¹. Therefore, $N_2O_5$ uptake was proposed to be dominantly responsible for the $NO_3$ loss and the initial s(t)  set to 1. Eq. 3 can describe the  sum concentration  of $NO_3$ and $N_2O_5$. Assuming the equilibrium between $NO_3$ and $N_2O_5$ is maintained after a period,  based on the temperature-dependent equilibrium rate constant ($k_{eq}$) and the modeled $NO_2$ at a certain time, Eq. 4 can be used to determine

the ratio of $N_2O_5$ to $NO_3$. Combined, Eqs. 1–4 allow for the calculation of  $NO_3$ and $N_2O_5$ concentrations considering stable $NO_3$ and $N_2O_5$ loss rate constants ($k_{NO3}$ and $k_{N2O5}$, respectively). In the second step, a new s(t) was calculated using the data from the first step (Eq. 5), new initial $NO_2$ and $O_3$ concentrations were then approximated, and $NO_3$ and $N_2O_5$ values were derived using the same method as used in the first step. This process was repeated until the difference between the two s(t) values was less than 0.005. The number of adjustments to a new s(t) could not be calculated more than 10 times. Otherwise, the calculating process would become non-convergent.

The modeled $N_2O_5$ concentrations and given $k_{N2O5}$ were then used to estimate $pNO_3^-$ formation. The $HNO_3$ produced in R4 was not considered because many of the products are organic nitrates (Brown and Stutz, 2012). Here $k_{NO3}$ and $k_{N2O5}$ denote the pseudo-first-order reaction rate constants of the total $NO_3$ reactivity caused by ambient VOCs and $N_2O_5$ heterogeneous uptake, respectively. $k_{N2O5}$ is given in Eq. 6. $S_a$ is  aerosol surface area, $C$ is the mean molecular speed of $N_2O_5$, and $\gamma_{N2O5}$ is the $N_2O_5$ uptake coefficient. Sunset and sunrise times during the measurements were 16:55 and 07:30 (Chinese National Standard Time, CNST), and the model was run from sunset to sunrise, with the running time set to 14.5 h.

$$\sim\text{(Eq. 1)}\quad \frac{d\,[O_3]}{dt} = -k_{\text{NO2+O3}}k_{NO2+O3}[O_3][NO_2] \tag{1}$$

$$\sim\text{(Eq. 2)}\quad \frac{d\,[NO_2]}{dt} = -2 \times k_{\text{NO2+O3}} = -(1+s(t)) \times k_{NO2+O3}[O_3][NO_2] \tag{2}$$

$$\sim\text{(Eq. 3)}\quad \frac{d\,[NO_3+N_2O_5]}{dt} = k_{\text{NO2+O3}}k_{NO2+O3}[O_3][NO_2] - k_{\text{N2O5}}k_{N2O5}[N_2O_5] - k_{\text{NO3}}
[revised manuscript text omitted]

---

## Author Response (AR2)

**We thank the editor and referees for his/her careful reading and constructive comments on our manuscript. As detailed below, the referee's comments is in black, our response to the comments is in blue. New or modified text is in red.**

**Referee #1**

The topic of the heterogeneous process of NOx is of interest to the community, and the vertical measurements of chemical composition in winter haze are particularly valuable. My main concern, however, on the present work is that the model was poorly constrained by observations and had to reply on too many assumptions. The vertical measurements only included O3, NO, and NO2 which enabled calculation of NO3 production, but several key parameters, such as N2O5, VOCs and aerosol surface area density, for loss of NO3, N2O5 production and subsequent loss to nitrate were not measured, making it very difficult, if not impossible, to evaluate nighttime reactions of NOx and for nitrate formation. In addition, the analysis was only based on one profile measured in the early evening, and too many assumptions in the model calculations were not well justified. All these make it very difficult to judge the validity of the conclusions drawn from the analysis. The authors are advised to carefully consider and reduce these uncertainties which could lead to large bias and possible errors.

The Referee's concerns can be summarized in two points as follows:
Point 1, there were limited set of parameters such as $NO$-$NO_2$-$O_3$, it may subject to large uncertainties when used for estimating the loss rate of $N_2O_5$.
Point 2, the representative of the case study for the winter Beijing.
We fully understand these concerns and we agree these are fair questions which needs to be addressed better.

For the point 1 concern, there will be large uncertainties of our method if it is summer, nevertheless, the winter Beijing conditions (low temperature and high $NO_2$) offered us big advantages in avoiding uncertainties. The advantages come from two folds, on one hand, the ratio of $N_2O_5$ to $NO_3$ was normally over 100:1 in the lower temperature and high $NO_2$ condition and thus the production rate of $NO_3$ was mostly in balance with the $N_2O_5$ loss rate (as shown in Fig. 7); on the other hand, the uncertainty of the iterative box model was controllable in polluted winter time as suggested by Wagner et al., (2013) and McDuffie et al., (2018).

For the point 2 concern, here we found a special case show the importance of the $N_2O_5$ uptake to the particulate nitrate formation in the urban canopy of Beijing which fits the recent discovery on the nighttime boundary layer diagnosis (Zheng et al., 2015; Zhong et al., 2017). Nevertheless, we acknowledged that more vertical profile measurements are required to elucidate the general characteristics of the chemical development of the nighttime chemistry in winter Beijing.

In response to the point 1 concern, we checked the uncertainty by many sensitivity tests of $k_{NO3}$ and $k_{N2O5}$. As for $k_{NO3}$, the influence was very small even set $k_{NO3}$ up to 0.02 s$^{-1}$. As for $k_{N2O5}$, the aerosol surface area was reasonably constrained. Although the uptake coefficient fixed to 0.005 without field data constrain, the sensitivity test showed that the $N_2O_5$ uptake processes were not limited by the $N_2O_5$ uptake coefficient (see Fig. 7). Overall, we think the uncertainties of the chemistry were well characterized in the revised manuscript of our previous response.

In response to the point 2 concern, we revised our abstract and conclusion as follows,

**Abstract:** "Modeling results show the specific case that the nighttime integrated production of $pNO_3^-$ for the high-altitude air mass above urban Beijing was estimated to be 50 µg m$^{-3}$ and enhanced the surface-layer $pNO_3^-$ the next morning by 28 µg m$^{-3}$ through vertical mixing. The overnight $NO_x$ loss via $NO_3$-$N_2O_5$ chemistry was efficient aloft (> 50%). The nocturnal $NO_x$ loss was maximized once the $N_2O_5$ uptake coefficient was over 2×10$^{-3}$ on polluted days with $S_a$ was 3000 µm$^2$ cm$^{-3}$ in wintertime. The case study provided a chance to highlight that $pNO_3^-$ formation via $N_2O_5$ heterogeneous hydrolysis may be an important source of the particulate nitrate in the urban airshed during wintertime."

**Conclusion:** "Our result emphasized the importance of the heterogeneous chemistry aloft the city through a case study. The model simulation also demonstrated that during the heavy PM pollution period, the particulate nitrate formation capacity via $N_2O_5$ uptake was easily maximized in the high altitude above urban Beijing, even with low $N_2O_5$ uptake coefficient."

**Referee #3**

General comments:

Kinetics and mechanisms of nitrate formation are of great concern in our society, and the manuscript has reported considerable Ox (O3+NO2) loss in the nocturnal residual layer in winter Beijing, immediately indicates potential nitrate formation in that nocturnal residual layer in winter Beijing, since the dominate sink of Ox is known to be nitrate formation. The authors also construct a model and run several sensitivity tests to illustrate that the N2O5 uptake is mainly responsible for the Ox loss and thus the inorganic nitrate formation stands out. Therefore, I would recommend publication of this manuscript on ACP on condition that specific weakness (see below) is resolved.

Specific comments:

1. The main premise for your analysis on the air mass evolution (or inorganic nitrate formation) overnight is that you have measured the same air mass in the evening measurement and the next morning measurement. However, dilution, exchange and mixing of the air mass cannot be ruled out considering the non-zero wind speed. How would the uncertainties originated from physical changes of the air mass affect your analysis and those many conclusions?

The uncertainty caused by the physical changes of the air masses were analyzed from two folds, one is the dilution and the other is the mixing and exchange of the air mass.

   With respect to the impact of the dilution process, it would decrease the mixing ratio of $NO_2$, $O_3$, $NO_3$ and $N_2O_5$, and leads to a lower contribution to the particulate nitrate formation. An additional loss process for trace gases with a lifetime of 24 h was assumed for calculated species in the sensitivity test (Lu et al., 2012). The result shows that the integrated production of particulate nitrate decreased 28% compared with base case.

   With respect to the exchange and mixing of the air mass at high altitude during nighttime in polluted winter, the stable atmospheric stratification was featured with strong inversion (Zhong et al., 2017). The nocturnal atmosphere is stable and layered, the upward mixing from the surface is minimized, and air masses above the surface are less affected by nocturnal emissions (Wagner et al., 2013). Nevertheless, the injection by warm combustion sources or the clean air mass can affect the air mass in fact. If the warm combustion source emitted NOx into the air mass after sunset, which would increase the mixing ratio of Ox, and restart the zero time of the model. Accounting for the uncertainties from the mixing, sensitivities tests of the box model to shorting the duration of 25%, the bias of the integrated $pNO_3^-$ throughout the night was small within 12% relative to base case. If the air mass was affected by the clean air mass from the north, it would be featured with very low NOx and about 40 ppbv $O_3$ (background condition), which was not consistent with our observation.

   Overall, the physical changes actually affect the reaction processes in the canopy of urban Beijing in winter time, and leads to a bias of the prediction of particulate nitrate formation. These uncertainties are further discussed in our revised manuscript.

We added a paragraph in the revise text to discuss the uncertainties originated by the physical changes of the air mass as: "The uncertainty caused by the physical changes of the air masses were analyzed from two folds, one is the dilution and the other is the mixing and exchange of the air mass. With respect to the impact of the dilution process, it would decrease the mixing ratio of $NO_2$, $O_3$, $NO_3$ and $N_2O_5$, and leads to a lower contribution to the particulate nitrate formation. An additional loss process for trace gases with a lifetime of 24 h was assumed for calculated species in the sensitivity test (Lu et al., 2012). The result shows that the integrated production of particulate nitrate decreased 28% compared with base case. With respect to the exchange and mixing of the air mass at high altitude during nighttime in polluted winter, the stable atmospheric stratification was featured with strong inversion (Zhong et al., 2017). The nocturnal atmosphere is stable and layered, the upward mixing from the surface is minimized, and air masses above the surface are less affected by nocturnal emissions (Wagner et al., 2013). Nevertheless, the injection by warm combustion sources or the clean air mass can affect the air mass in fact. If the warm combustion source emitted NOx into the air mass after sunset, which would increase the mixing ratio of Ox, and restart the zero time of the model. Accounting for the uncertainties from the mixing, sensitivities tests of the box model to shorting the duration of 25%, the bias of the integrated $pNO_3^-$ throughout the night was small within 12% relative to base case. If the air mass was affected by the clean air mass from the north, it would be featured with very low NOx and about 40 ppbv $O_3$ (background condition), which was not consistent with our observation."

2. If the story of Ox loss and inorganic nitrate formation in the nocturnal residual layer in winter Beijing in one of your measurements is credible regardless of question #1, I did not see why you could safely extrapolate the story of the specific case study to (1). the winter nights since no gradient of NO, NO2 and O3 has been spotted in your measurements. (2) general situation of Beijing (3) the 50 μg m$^{-3}$ contribution due to (1) and (2), and also that the structure of nocturnal boundary layer is not characterized anywhere in your study.

Actually, it's difficult to safely extrapolate our story to all the general situation of Beijing. Our specific study described an upper limit case about the capacity of the particulate nitrate produced by $N_2O_5$ uptake during nighttime and contribute to the following daytime PM mass concentration in winter Beijing.

Firstly, with respect to the case that NO concentration within 240 m was high (e.g., December 20, 2016), we did not have the vertical profile of these species above 240 m. But it is possible that we did not reach to the residual layer due to the nocturnal boundary layer was higher than 240 m under this condition.

Secondly, several studies showed that the nocturnal boundary layer was lower than 200

m in the polluted days in winter Beijing, suggested our specific case was general in Beijing (Zheng et al., 2015; Zhong et al., 2017).

Thirdly, the 50 µg m$^{-3}$ contribution in the case study was not so general, as the mixing ratios of the precursors are different in different polluted episodes. But it highlighted that N$_2$O$_5$ heterogeneous uptake in the residual layer where far away from the NO emission would have a significant contribution to the particulate nitrate formation.

Overall, our case study just provide a chance to qualitative look insight the importance of N$_2$O$_5$ uptake to PM pollution in vertical scale, and more field studies was need to quantify the contribution. Here we modest our statement in the abstract and conclusion as following:

**Abstract:** "Modeling results show the specific case that the nighttime integrated production of pNO$_3^-$ for the high-altitude air mass above urban Beijing was estimated to be 50 µg m$^{-3}$ and enhanced the surface-layer pNO$_3^-$ the next morning by 28 µg m$^{-3}$ through vertical mixing. The overnight NO$_x$ loss via NO$_3$-N$_2$O$_5$ chemistry was efficient aloft (> 50%). The nocturnal NO$_x$ loss was maximized once the N$_2$O$_5$ uptake coefficient was over 2×10$^{-3}$ on polluted days with $S_a$ was 3000 µm$^2$ cm$^{-3}$ in wintertime. The case study provided a chance to highlight that pNO$_3^-$ formation via N$_2$O$_5$ heterogeneous hydrolysis may be an important source of the particulate nitrate in the urban airshed during wintertime."

**Conclusion:** "Our result emphasized the importance of the heterogeneous chemistry aloft the city through a case study. The model simulation also demonstrated that during the heavy PM pollution period, the particulate nitrate formation capacity via N$_2$O$_5$ uptake was easily maximized in the high altitude above urban Beijing, even with low N$_2$O$_5$ uptake coefficient."

3. Constraint or comparison of your model to your measurements of Ox would help.
Thanks for the suggestion, we added the description about the observed and modelled Ox loss as following:

"The pNO$_3^-$ formation by N$_2$O$_5$ heterogeneous uptake from sunset to the measurement time can be calculated using Eq. 7, which was significant of 24 µg m$^{-3}$ after sunset above 150 m. The particulate nitrate formed in 4.5 hours was equivalent to 13 ppbv O$_x$ loss and consistent with the observation (15 ppbv) (Fig. 6b). Where the 1.5:1 relationship between O$_x$ and pNO$_3^-$ was used to calculate the O$_x$ equivalence (S. S. Brown et al., 2006). "

"The box model enabled the analysis of the integrated pNO$_3^-$ and ClNO$_2$ via N$_2$O$_5$ uptake throughout the night. As shown in Fig. 6c, the modeled integrated pNO$_3^-$ went as high as 50 µg m$^{-3}$. The integrated pNO$_3^-$ at sunrise was equal to the loss of 27 ppbv O$_x$, showing a good agreement with the observed O$_x$ missing (25 ppbv) aloft in the morning hours."

4. Need great improvement in English writing. Here only the comments on the first page (line1-line31) of the manuscript are listed.
Thanks for the detailed English gramma correction, we went through the text and improved the language accordingly.

Line 1: consider "fast/rapid formation" or "big formation rate"
Changed to: "fast particulate nitrate formation"

Line 2: define "chemical reactive layer"
Changed the "chemical reactive layer" to "aloft".

Change "during wintertime in Beijing" to "in winter Beijing"
Changed accordingly.

Line 13: is not "dominant"
Changed to "an important component".

Line 16: measurement of what?
Simultaneous ground-based and tower-based measurements of $NO_x$ and $O_3$

Line 18: change to "due to N2O5 concentration of near zero controlling by high NO emission and NO concentration"
Changed accordingly.

Line19: No will not "limiting the production of N2O5"
Deleted the statement.

Larger or large, higher or high (if being specific on altitude like > 150 m, just delete "high")
Changed to: "In contrast, the contribution from $N_2O_5$ uptake was larger at higher altitudes (e.g., > 150 m)".

Line 20: Large or larger (Large is not a proper adj. for formation or missing)
Changed to "large".

Line 21: define "production potential"
The "production potential" changed to "integrated production".

Line 22: higher or high
Changed to "high".

Line 23: delete "significantly"
Deleted accordingly.

Line 24-27: a conclusion drawn from nowhere, hard to follow

Deleted these words.

Line 28:higher or high

Changed to "high".

Line 30: define "reactive air masses"

Deleted the sentence.

Line 28-31: "haze formation", "formation and development of the reactive air masses" is not the topic of this study. Also, be specific on how are you going to improve the "chemical-transport model" based on the contribution of your study?

Deleted the sentence.

**Referee #4**

General comments:

As noted by the other reviewers of this manuscript, the paper presents new results on the formation of nitrate aerosol by N2O5 uptake at modest altitudes above surface level during winter in Beijing. The result is important since ozone titration by NOx at surface level makes the production rate at the surface zero. The paper quantifies the production rate in an altitude range up to 250 m above the surface and is thus a valuable contribution to the literature.

The authors quantify uncertainties in their analysis due to NO3 reactivity and ClNO2 production, both of which are unmeasured. They do not quantify the uncertainty due to partitioning between gas phase HNO3 and particulate nitrate. This aspect should be addressed, even if it is simply to state by way of assumption that all nitrate formed from N2O5 uptake goes to the particle phase rather than the gas phase. If the authors have other information to indicate that partitioning (other literature) they should state this explicitly.

The partition of gas phase $HNO_3$ and particulate nitrate is an important aspect affected the contribution of $N_2O_5$ uptake producing particulate nitrate. We assumed that all nitrate formed from $N_2O_5$ uptake goes to the particle phase, the assumption would lead to the $N_2O_5$ uptake had an upper limit particle phase nitrate contribution.

While during the wintertime in Beijing, the mixing ratio of $NH_3$ was rich to tens of ppbv and always much higher than the nocturnal gas phase $HNO_3$ (e.g., Liu et al., 2017). The high $NH_3$ suppressed the degassing of particulate nitrate effectively. The measurement of gas phase $HNO_3$ and particulate nitrate in the surface layer of Beijing showed the soluble nitrate favor to particle phase in winter, especially in polluted days. For example, the nocturnal ratio of particulate nitrate to total soluble nitrate was larger than 0.95 on average (Liu et al., 2017; unpublished data from the BEST-ONE campaign in Beijing (Tan et al., (2018)). Due to the low temperature and high RH in high altitude, the ratio would increase and the degassing of particulate nitrate is small.

Therefore, the assumption that all the $N_2O_5$ uptake contributes to the particle phase is reasonable in the wintertime in the canopy of Beijing.

Adding a paragraph as: "In the calculation of the particulate nitration formation by $N_2O_5$ uptake, an assumption is that all nitrate formed from $N_2O_5$ uptake goes to the particle phase rather than the gas phase. The assumption would lead to an upper bias as the degassing of gas phase $HNO_3$ from particulate nitrate. While during the wintertime in Beijing, the mixing ratio of $NH_3$ was rich to tens of ppbv and always much higher than the nocturnal gas phase $HNO_3$ (e.g., Liu et al., 2017). The high $NH_3$ suppressed the degassing of particulate nitrate effectively. The measurement of gas phase $HNO_3$ and particulate nitrate in the surface layer of Beijing showed the soluble nitrate favor to particle phase in winter, especially in polluted days. For example, the nocturnal ratio of particulate nitrate to total soluble nitrate was larger than 0.95 on average (Liu et al., 2017). Due to the low temperature and high RH in high altitude, the ratio would increase and the degassing of particulate nitrate is negligible."

Otherwise the manuscript should be published subject to the following minor comments and grammatical corrections.

Specific comments:

Line 26: remove the word "easily"
Removed accordingly.

Line 27: The result for gamma(N2O5) is specific to the very large aerosol surface area present in Beijing during these events. The sentence in the abstract should note this so as not to imply that a gamma value of 2x10-3 is generally the point at which other regions would become insensitive to this parameter.
Thanks for the suggestion, we added the Sa information in this sentence as following:
"The nocturnal $NO_x$ loss was maximized once the $N_2O_5$ uptake coefficient was over $2\times10^{-3}$ on polluted days with $S_a$ was 3000 $\mu m^2$ $cm^{-3}$ in wintertime."

Line 28: Suggest replacing "could be" with "is". At the author's discretion.
Changed accordingly.

Line 43: replace "severely limited" with "very low in concentration"

Changed accordingly.

Line 162 and equation 5: The idea behind the definition of s(t) is fairly clear, but the form of equation 5 is not. Further explanation of the form of the equation is required.

The s(t) is between 0 and 1 and expressed as Eq. 5, the physical meaning s(t) is the ratio of $NO_3$ production which goes through $N_2O_5$ (either as $N_2O_5$ or lost through uptake) to the total $NO_3$ production (Wagner et al., 2013)

Figure 2: The scale for O3 in the upper panel goes to 100 ppbv, while the O3 itself only goes to 25 ppbv. The scale should show the actual variability in O3.

The scale of $O_3$ in Figure 2 changed to 40 ppb.

Line 235: The choice of kNO3 is arbitrary and is intended to simply represent a high value. The word arbitrary should appear in the sentence, i.e., "… kNO3 was set to an arbitrary and relatively high value of …"

Changed accordingly.

Line 268: Do the authors mean to refer to December 19 rather than December 18?

Changed to: "The vertical profile on December 19 was different with that on December 20."

Line 277: Is NO greatly diminished, or zero? Nonzero O3 at night implies zero NO if the mixing ratio of O3 is sustained for any length of time.

Yes, The NO is zero and changed as: "corresponding to zero NO concentration".

Line 303: Omit the word "about"

Changed accordingly.

Line 306: "rapid" in place of "quick"

Changed accordingly.

Line 314: What relationship between Ox and pNO3- has been used to calculate the Ox equivalence in Figure 6? Have the authors assumed a 1:1 relationship, or have they used the Ox equivalent in pNO3, which is larger than 1? See: Brown et al., Nocturnal odd-oxygen budget and its implications for ozone loss in the lower troposphere. Geophys. Res. Lett., 2006. 33: p. L08801.

The 1.5:1 relationship was used to calculate the equivalence of $O_x$ and $pNO_3^-$. Which is the same as the reference recommend.

Added a sentence as: "The 1.5:1 relationship between $O_x$ and $pNO_3^-$ was used to calculate the Ox equivalence (S. S. Brown et al., 2006)."

Line 332: Can the authors compare the 28 micrograms m-3 figure to the day over day change in nitrate mass during haze events in Beijing? In other words, what is the daily growth in nitrate mass during either this event or during typical events, and how much is explained by this 28 microgram m-3 per night rate?

The enhancement of particulate nitrate of 28 $\mu g\ m^{-3}$ was a high contribution case, but the case with large enhancement of particulate nitrate of ~20 $\mu g\ m^{-3}$ can be found at the same site on the tower in polluted winter Beijing (c.a. figure 4(a) from Sun et al., (2015)). The fast growth of particulate nitrate with 50 $\mu g\ m^{-3}$ per day was found in BEST-ONE campaign in winter Beijing, 2016. The quantitative particulate nitrate enhancement by $N_2O_5$ uptake was case by case, and the result represented by our specific case just address the significance of $N_2O_5$ uptake in the canopy of Beijing.

Line 334: To what feature are the authors referring in stating a morning peak of 60 micrograms $m^{-3}$ on Dec 20? This feature is not apparent in Figure 1.

The feature is shown in Figure 2(a) and colored as red line.

346: Correct English grammar. Use a period and new sentence rather than a comma. The second part of the sentence should read: "Low N2O5 uptake coefficients correspond to several types of aerosols, such as …"

Thanks for the suggestion, we changed accordingly.

Line 352: Logic of sentence is incorrect. The ClNO2 yield is not the variable that maximizes the conversion capacity of N2O5, as the sentence implies. Rephrase as: "The conversion capacity of N2O5 uptake to pNO3 is maximized for a given, fixed value of the ClNO2 yield"

Changed accordingly.

Line 363: It is not clear what is intended by the phrase "valid NOx loss." The authors should clarify or search for other wording.

Here we use: "NOx removal".

Line 364: "the N2O5 uptake coefficient" rather than "N2O5 uptake"

Changed accordingly.

Line 369: remove the word "was"

Changed accordingly.

Line 373: "become insensitive to $\gamma$(N2O5)." Then start a new sentence "This region is defined as …"

Changed accordingly.

Line 387: Sentence needs improved English grammar. The meaning of "during the heating period" is not clear. Does this refer simply to colder weather during the winter season?

The heating period means very cold period in winter Beijing, and the government would supply the heating water from the thermal power plant.

Line 394: This result of 2.5 ppbv refers to a model, not a measured value. This should be made clear.

Revised as: "The modelled formation of $ClNO_2$ aloft throughout the night reached 2.5 ppbv,"

Line 397: "As the error of pNO3 formation simulation was subject to" should be replace by "Since the modeled pNO3 formation is sensitive to"

Changed accordingly.

Line 399: The reference is to Figure 8, not Figure 7.

Corrected accordingly.

Line 415: replace "evidenced" with "found evidence for"

Changed accordingly.

Line 718, figure caption 7: "via N2O5 uptake" rather than "on N2O5 uptake", "NO2 and O3 were set to", "Sa was set", "reaction time was set.

Changed accordingly.

[revised manuscript text omitted]